# Transient nicotine exposure in early adolescent male mice freezes their dopamine circuits in an immature state

Lauren M. Reynolds [1,2] ✉, Aylin Gulmez[1], Sophie L. Fayad[1,2],
Renan Costa Campos [3], Daiana Rigoni [3], Claire Nguyen [2],
Tinaïg Le Borgne[1,2], Thomas Topilko[4], Domitille Rajot[4], Clara Franco [2],
Sebastian P. Fernandez[3], Fabio Marti [1,2], Nicolas Heck [2],
Alexandre Mourot [1,2], Nicolas Renier [4], Jacques Barik [3] &
Philippe Faure [1,2] ✉

How nicotine acts on developing neurocircuitry in adolescence to promote later addiction vulnerability remains largely unknown, but may hold the key for informing more effective intervention efforts. We found transient nicotine exposure in early adolescent (PND 21-28) male mice was sufficient to produce a marked vulnerability to nicotine in adulthood (PND 60 +), associated with disrupted functional connectivity in dopaminergic circuits. These mice showed persistent adolescent-like behavioral and physiological responses to nicotine, suggesting that nicotine exposure in adolescence prolongs an immature, imbalanced state in the function of these circuits. Chemogenetically resetting the balance between the underlying dopamine circuits unmasked the mature behavioral response to acute nicotine in adolescent-exposed mice. Together, our results suggest that the perseverance of a developmental imbalance between dopamine pathways may alter vulnerability profiles for later dopamine-dependent psychopathologies.

Smoking is a major contributor to disease burden worldwide, driven by addiction to nicotine[1]. The prognosis is particularly bleak for the up to 90% of adult smokers who began in adolescence[2], as early onset drug use is associated with an elevated risk of addiction throughout the lifetime[3]. Indeed, epidemiological studies associate adolescent nicotine use with increased risk of not only longer and heavier smoking careers[4], but also with the development of anxiety disorders and depression[5,6], and with the problematic consumption of other drugs, such as alcohol[4]. Studies in animal models have focused to date largely on the differences in the rewarding effects of nicotine in adolescent vs adult rats and mice and on how adolescent nicotine exposure can influence later measures of nicotine reward[7,8], but less is known about

how nicotine impacts the adolescent development of neural circuitry to produce these enduring outcomes.

Mature cognitive, emotional, and motivational behaviors emerge across adolescence in parallel to the development of their underlying neurocircuitry. Dopamine (DA) circuitry, in particular, is increasingly considered as a "plasticity system" where its structure and function is shaped by experience during development, creating adaptive behavioral profiles that can endure throughout the lifetime[9,10]. While this window may allow for performance optimization in the context of a specific environment, it also likely demarcates a period of increased vulnerability to environmental insult[11]. Nicotine acts directly on DA neurons through its action on nAChRs[12], a family of pentameric

[1]Plasticité du Cerveau CNRS UMR8249, École supérieure de physique et de chimie industrielles de la Ville de Paris (ESPCI Paris), Paris, France. [2]Neuroscience Paris Seine CNRS UMR 8246 INSERM U1130, Institut de Biologie Paris Seine, Sorbonne Université, Paris, France. [3]Université Côte d'Azur, Nice 06560, France; Institut de Pharmacologie Moléculaire & Cellulaire, CNRS, UMR7275 Valbonne, France. [4]Laboratoire de Plasticité Structurale INSERM U1127, CNRS UMR7225, Sorbonne Université, ICM Institut du Cerveau et de la Moelle Epinière, Paris, France. ✉e-mail: laurenreynoldsm@gmail.com; phfaure@gmail.com

ligand-gated ion channels. While the effects of nicotine on DA neurons are best studied in adult animals[13–15], some evidence does suggest that both the immediate and enduring effects of nicotine on ventral tegmental area (VTA) DA neurons differ between adolescent and adult administration[16,17]. However, these studies treat DA neurons as a single, homogenous population; whereas VTA DA neurons are increasingly recognized to belong to anatomically distinct circuits, to possess diverse molecular signatures, and to differ in their responses to external stimuli[18–22]. Indeed, nicotine simultaneously produces both reinforcing and anxiogenic behavioral effects[23,24], which rely on distinct DA pathways: the activation of DA neurons projecting to the nucleus accumbens (NAc) produces reinforcement, whereas the inhibition of amygdala (AMG)-projecting DA neurons produces anxiety-like behavior[23]. Dopaminergic axons are still connecting to forebrain targets in adolescence[11], and are sensitive to disruption by other psychostimulant drugs of abuse[25]; however whether and how nicotine may alter the development of distinct DA circuits in adolescence to drive later addiction vulnerability remains unknown. Here, we found that male mice exposed to nicotine during early adolescence showed persistent behavioral and electrophysiological markers of vulnerability to nicotine re-exposure as adults, which was associated with a restructuring of dopaminergic functional connectivity as to create an imbalance in VTA-NAc and VTA-AMG signaling. Interestingly, the behavioral and electrophysiological responses to nicotine in these adult mice exposed to nicotine as adolescents closely resembled the phenotype of naïve adolescent mice, suggesting this signaling imbalance is a marker of immaturity in dopaminergic circuits. Finally, we show that chemogenetically resetting an adult-like balance in dopamine signaling in response to nicotine restores a mature behavioral response in adolescent-exposed mice, causally implicating the development of this pathway in a type of naturally occurring resilience to nicotine use that develops in unexposed adults.

## Results

### Exposure to nicotine during adolescence, but not during adulthood, enduringly increases vulnerability to nicotine use

To model nicotine exposure in the period when human adolescents are most likely to start smoking or using e-cigarettes[26,27], we focused on comparing exposure to nicotine early in adolescence in mice (~PND21 – PND 28)[11] with exposure in adulthood (>PND60). To this end, male mice underwent a one-week exposure to nicotine (NIC, 100 μg/ml in 2% saccharin) or to 2% saccharin only (SAC) in their home cage drinking bottle either in early adolescence or in adulthood. All mice were tested for sucrose and nicotine consumption five weeks later in a continuous-access free-choice oral self-administration paradigm (Fig. 1A). Mouse weight stayed stable over the course of the task (Supplementary Fig. 1A), and their drinking behavior followed their diurnal behavioral profile, with the greatest drinking volumes shortly after the beginning of the dark cycle, regardless of the solution presented (Supplementary Fig. 1B). All mice showed a strong, dose-dependent preference for the sucrose-containing solution over water (Fig. 1B left, 1 C left, Table 1A, C), with similar overall volumes of liquid consumed (Supplementary Fig. 1C, D), regardless of their pretreatment solution or age.

While rodents, and mice in particular, do not acquire intra-venous self-administration of nicotine as readily as with other stimulant drugs of abuse[28], we found that mice voluntarily self-administered nicotine at all doses when tested in an oral nicotine intake paradigm (Fig. 1B, C), with a strong effect of dose across all experiments indicative of a titration effect[29,30]. Mice showed the greatest preference for the nicotine solution at the 10 μg/ml concentration, where it represented 65-80% of their liquid intake. Adult mice exposed to NIC in adolescence consumed more nicotine solution as a percentage of their total liquid intake than their congeners exposed only to SAC in adolescence at all doses tested, (Fig. 1B middle, Table 1B), with differences in the daily

dose of nicotine most obvious at the 50 μg/ml and 100 μg/ml doses. This did not result from side bias in consumption, as all mice were able to track the position of the nicotine or sucrose solution (Supplementary Fig. 1G). In stark contrast, mice exposed to NIC in adulthood did not show different consumption behavior from their SAC-treated counterparts (Fig. 1C, Table 1E), suggesting that this 1-week pretreatment regimen is too mild to produce enduring changes to the adult brain. Liquid intake across the nicotine sessions was stable across all groups of mice, and only the mice pretreated with NIC in adolescence showed a significant difference from their SAC-treated counterparts in the volume (mL) of nicotine solution consumed (Supplementary Fig. 1D).

Nicotine has both reinforcing and anxiogenic properties[23], both of which have been hypothesized to contribute to addiction liability. We thus next investigated how exposure to NIC in adolescence affects the anxiogenic properties of acute nicotine delivery in the elevated O-Maze (EOM, Fig. 1D), or the reinforcing effects of nicotine in a conditioned place preference paradigm (CPP, Fig. 1D). We have previously found that naïve adult male mice show a pronounced, time-dependent reduction of time spent in the open arms of the EOM following nicotine administration[23]. Here, we found that mice exposed to SAC in adolescence resembled our results in naïve mice: these mice showed an anxiogenic response to nicotine, as evidenced by a significant reduction in time spent in the open arms in the 6-to-9-minute block when compared to saline-injected counterparts. There was, however, no difference in time spent in the open arms in the 6-to-9-min block between saline and nicotine injected mice that were previously exposed to nicotine in adolescence, indicating that exposure to NIC in adolescence abolished this anxiogenic effect (Fig. 1E, Table 1G). However, the anxiogenic effect of acute nicotine injection was preserved in mice exposed to SAC or to NIC pretreatment as adults (Fig. 1H, Table 1I). Adult mice exposed to NIC in adolescence, but not those exposed in adulthood, were also impervious to reductions in open arm entries (Sup Fig. 1I) and suppression of locomotor activity (Supplementary Fig. 1I) in response to acute nicotine injection.

We have previously shown that adult male mice show CPP to a 0.5 mg/kg dose of nicotine[31], but not to a 0.2 mg/kg dose of nicotine[32]. Here, we tested whether adult mice treated with nicotine in adolescence were more sensitive to the reinforcing effects of a low dose of nicotine. Mice exposed to SAC in adolescence did not show CPP to a 0.2 mg/kg dose of nicotine, in accordance with our previous results in naïve adult mice. However, mice treated with NIC in adolescence showed a significant preference for the nicotine-paired chamber at this same dose (Fig. 1F, Table 1H), suggesting that these mice are more sensitive to the reinforcing effects of nicotine. Mice treated with NIC or with SAC as adults showed no preference for the nicotine-paired chamber (Fig. 1I, Table 1J).

Together, these results define an adolescent period where exposure to nicotine produces an enduring profile of altered response to later nicotine, featuring a reduction of its anxiogenic properties, an increased sensitivity to its reinforcing properties, and an increase in voluntary consumption consistent with a vulnerable phenotype. This finding further suggests vulnerability to nicotine is titrated along an age spectrum, where ongoing developmental processes create resilience to the drug as adolescents progress into adulthood.

### Nicotine in adolescence restructures nicotine-responsive networks in the adult brain

To dissect how this nicotine treatment in adolescence alters the functional organization of brain networks, we performed whole-brain activity mapping in a subset of mice that underwent testing in the EOM (Fig. 2A). Mice treated with NIC or SAC in adolescence or adulthood received an acute injection of nicotine or saline, were tested for anxiety-like behavior in the EOM, and were perfused ~1 h later at peak cFos expression. Their brains were then cleared using iDISCO,

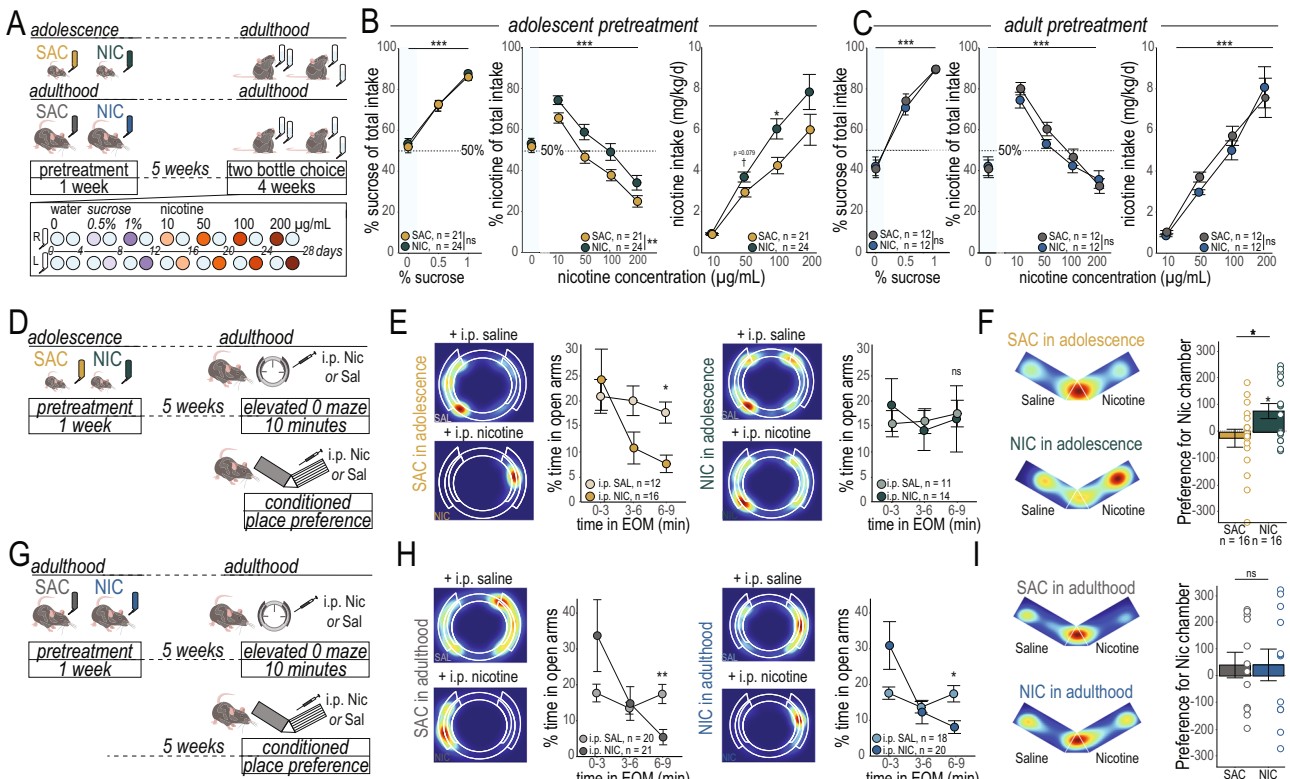

**Fig. 1 | Brief exposure to nicotine in adolescence induces a life-long imbalance between the rewarding and anxiogenic effects of later exposure. A** *Top:* Experimental timeline. *Bottom:* Timeline for the oral free-choice self-administration paradigm. **B** Adult mice exposed to NIC in adolescence (*left*) show equivalent preference for a sucrose solution as their SAC-treated counterparts. Adult mice exposed to NIC in adolescence showed a higher percentage intake of the nicotine-containing solution over all treatment doses (*center*). NIC-pretreated mice self-administered a higher daily dose of nicotine, with a significant difference at the 100 µg/ml dose (*right*, Table 1A–C). **C** Adult mice exposed to NIC in adulthood (*left*) show equivalent preference for a sucrose solution as their SAC-treated counterparts. Mice pretreated with NIC in adulthood also did not differ from controls in the percentage of nicotine solution consumed (*center*), nor by their daily dose of nicotine (*right*, Table 1D–F). **D** Experimental timeline. **E** The anxiogenic properties of an acute nicotine injection are maintained in mice pretreated with SAC in adolescence (*left*) but blocked in adult mice that were treated with NIC in adolescence

(*right*, Table 1G). Graphs are separated by pre-treatment group for clarity, all statistical analyses were conducted on all four treatment conditions. **F** Adult mice treated with SAC in adolescence did not show CPP to a 0.2 mg/kg dose of nicotine. Adult mice treated with NIC in adolescence, however, showed CPP to this low dose of nicotine (Table 1H). **G** Experimental timeline. **H** Mice treated with NIC in adulthood show an equivalent anxiety-like response to acute nicotine (*right*) as their SAC-treated counterparts (*left*, Table 1I). Graphs are separated by pre-treatment group for clarity, but statistical analyses compared all four treatment conditions. **I** Mice treated with NIC or SAC as adults do not show CPP to a low dose of nicotine (Table 1J). All line graphs are presented as mean values ± SEM. Heatmaps are from representative individual animals. †$p < 0.08$, *$p < 0.05$, **$p < 0.01$, ***$p < 0.01$, ns = not significant. All statistical comparisons were two-sided. Holm's sequential Bonferroni corrections were used to correct for multiple comparisons. Detailed information about statistical testing is available in Table 1. Source data are provided as a Source Data file.

immunostained for cFos, and imaged on a lightsheet microscope[33]. Scans were automatically registered to the Allen Brain Atlas and cFos positive cells were quantified using the ClearMap analysis pipeline[34]. Adult mice exposed to NIC in adolescence had overall greater brain-wide activation in response to an acute injection of nicotine in the EOM, with a significant increase over SAC treated counterparts in 129 regions (Fig. 2B *left*). This included significant increases in cFos positive neurons in regions associated with addiction and addiction vulnerability, such as the nucleus accumbens (NAc), the prelimbic prefrontal cortex, and the pallidum; as well as in parts of the amygdala associated with anxiety and anxiety-like behavior (Anterior and Medial Amygdala). In contrast, mice that received NIC treatment as adults did not have major differences in cFos+ neuron expression from their SAC-treated counterparts (Fig. 2B *right*); nor were major differences seen between any groups in response to saline injections (Supplementary Fig. 2A, B).

To perform a more in-depth network analysis, we first identified a list of 36 brain regions implicated in anxiety behavior and nicotine response[35–39]. Correlation matrices show that nicotine-responsive regions are highly correlated in SAC-pretreated animals and can be organized into 4 modules (Fig. 2C *left*). When the same relationships

between these regions are probed in NIC pretreated mice, a global disruption of correlated activity is apparent within these modules (Fig. 2C *right*). Functional correlations between these regions were less striking in mice of any pretreatment condition that received saline injections before entering the EOM (Supplementary Fig. 2C, D), and mice that received SAC or NIC as adults had similarly organized functional connectivity (Supplementary Fig. 2E).

To investigate these relationships further, we next made network graphs from these nicotine-response correlation matrices, where each region was used as a node and each correlation as an edge, and we used Louvain community detection to organize these graphs into communities (Supplementary Fig. 2G)[40]. We found that while the overall network structure, including community members and inter-community edges, were similar between the two pretreatment groups, there was a significant reorganization of the functional relationships between the NAc, the VTA, and the basomedial and basolateral regions of the amygdala (Fig. 2C *bottom*). We then confirmed that these areas showed significant voxel-by-voxel changes in the number of cFos+ cells in grouped comparisons, with more active cells seen in the NAc and VTA of NIC pretreated mice, and fewer active cells in the medial Amg of NIC pretreated mice in response to nicotine re-injection (Fig. 2D). Mice that

## Table 1 | Detailed Statistics for Fig. 1

| N | Test | Factor | Statistic | Statistic Value | p value | Corresponding Figure |
|---|---|---|---|---|---|---|
| **A - % sucrose preference - adolescent pretreatment** | | | | | | |
| SAC = 21, NIC = 24 | Shapiro-Wilk normality test | | W | 0.9478 | 0.0001 | Fig. 1B left |
| | Wilcoxon rank sum test with continuity correction | SAC vs NIC at 0% | W | 261.0000 | 1.0000* | |
| | Wilcoxon rank sum test with continuity correction | SAC vs NIC at 0.5% | W | 256.0000 | 1.0000* | |
| | Wilcoxon rank sum test with continuity correction | SAC vs NIC at 1% | W | 297.0000 | 0.9340* | |
| **B - % nicotine preference - adolescent pretreatment** | | | | | | |
| SAC = 21, NIC = 24 | Shapiro-Wilk normality test | | W | 0.9925 | 0.3077 | Fig. 1B center |
| | Two-way mixed ANOVA | Interaction | F | 1.1080 | 0.3544 | |
| | | Treatment (between subj) | F | 8.4491 | 0.0058 | |
| | | Nic conc (within subjects) | F | 68.2984 | 0.0000 | |
| **C - nicotine intake (mg/kg/day) - adolescent pretreatment** | | | | | | |
| SAC = 21, NIC = 24 | Shapiro-Wilk normality test | | W | 0.8565 | 0.0000 | Fig. 1B right |
| | Wilcoxon rank sum test with continuity correction | SAC vs NIC at 10 µg/mL | W | 289.0000 | 0.4063* | |
| | Wilcoxon rank sum test with continuity correction | SAC vs NIC at 50 µg/mL | W | 350.0000 | 0.0796* | |
| | Wilcoxon rank sum test with continuity correction | SAC vs NIC at 100 µg/mL | W | 384.0000 | 0.0111* | |
| | Wilcoxon rank sum test with continuity correction | SAC vs NIC at 100 µg/mL | W | 334.0000 | 0.1274* | |
| **D - % sucrose preference - adult pretreatment** | | | | | | |
| SAC = 12, NIC = 12 | Shapiro-Wilk normality test | | W | 0.8997 | 0.0000 | Fig. 1C left |
| | Wilcoxon rank sum test with continuity correction | SAC vs NIC at 0% | W | 73.0000 | 1.0000* | |
| | Wilcoxon rank sum test with continuity correction | SAC vs NIC at 0.5% | W | 61.0000 | 1.0000* | |
| | Wilcoxon rank sum test with continuity correction | SAC vs NIC at 1% | W | 69.0000 | 1.0000* | |
| **E - % nicotine preference - adult pretreatment** | | | | | | |
| SAC = 12, NIC = 12 | Shapiro-Wilk normality test | | W | 0.9803 | 0.0755 | Fig. 1C center |
| | Two-way mixed ANOVA | Interaction | F | 1.0332 | 0.3947 | |
| | | Treatment (between subj) | F | 0.5357 | 0.4720 | |
| | | Nic conc (within subjects) | F | 57.9616 | 0.0000 | |
| **F - nicotine intake (mg/kg/day) - adult pretreatment** | | | | | | |
| SAC = 12, NIC = 12 | Shapiro-Wilk normality test | | W | 0.8945 | 0.0000 | Fig. 1C right |
| | Wilcoxon rank sum test with continuity correction | SAC vs NIC at 10 µg/mL | W | 46.0000 | 0.4229* | |
| | Wilcoxon rank sum test with continuity correction | SAC vs NIC at 50 µg/mL | W | 31.0000 | 0.0775* | |
| | Wilcoxon rank sum test with continuity correction | SAC vs NIC at 100 µg/mL | W | 56.0000 | 0.7417* | |
| | Wilcoxon rank sum test with continuity correction | SAC vs NIC at 100 µg/mL | W | 71.0000 | 0.9770* | |
| **G - % time in open arms of EOM - adolescent pretreatment** | | | | | | |
| SAC + SAL = 12, SAC + NIC = 16, NIC + SAL = 11, NIC + NIC = 14 | Shapiro-Wilk normality test | | W | 0.8339 | 0.0000 | Fig. 1E |
| | Kruskal-Wallis rank sum test | All groups, 0–3 min | Kruskal-Wallis chi-squared | 0.8896 | 0.8279 | |
| | Kruskal-Wallis rank sum test | All groups, 3–6 min | Kruskal-Wallis chi-squared | 6.1815 | 0.1031 | |
| | Kruskal-Wallis rank sum test | All groups, 6–9 min | Kruskal-Wallis chi-squared | 14.4753 | 0.0023 | |
| | Wilcoxon rank sum test with continuity correction | NIC-SAL vs NIC-NIC, 6–9 min | W | 46.0000 | 0.2847* | |
| | Wilcoxon rank sum test with continuity correction | SAC-SAL vs SAC-NIC, 6–9 min | W | 27.0000 | 0.0058* | |
| | Wilcoxon rank sum test with continuity correction | SAC-SAL vs NIC-SAL, 6–9 min | W | 62.0000 | 0.8294* | |

**Table 1 (continued) | Detailed Statistics for Fig. 1**

| N | Test | Factor | Statistic | Statistic Value | p value | Corresponding Figure |
|---|---|---|---|---|---|---|
| **H - CPP to 0.2 mg/kg nicotine - adolescent pretreatment** | | | | | | |
| SAC = 16, NIC = 16 | Wilcoxon rank sum test with continuity correction | SAC-NIC vs NIC-NIC, 6–9 min | W | 146.5000 | 0.3117* | Fig. 1F |
| | Shapiro-Wilk normality test | | W | 0.9628 | 0.3264 | |
| | One Sample t test | SAC difference from 0 | t | −0.6684 | 0.5140 | |
| | One Sample t test | NIC difference from 0 | t | 2.8120 | 0.0131 | |
| | Welch Two Sample t-test | SAC vs NIC | t | 2.3183 | 0.0277 | |
| **I - % time in open arms of EOM - adult pretreatment** | | | | | | |
| SAC + SAL = 20, SAC + NIC = 21, | Shapiro-Wilk normality test | | W | 0.7442 | 0.0000 | Fig. 1H |
| NIC + SAL =18, NIC + NIC = 20 | Kruskal-Wallis rank sum test | All groups, 0–3 min | Kruskal-Wallis chi-squared | 1.0547 | 0.7880 | |
| | Kruskal-Wallis rank sum test | All groups, 3–6 min | Kruskal-Wallis chi-squared | 4.5444 | 0.2084 | |
| | Kruskal-Wallis rank sum test | All groups, 6–9 min | Kruskal-Wallis chi-squared | 18.3625 | 0.0004 | |
| | Wilcoxon rank sum test with continuity correction | NIC-SAL vs NIC-NIC, 6–9 min | W | 91.0000 | 0.0284* | |
| | Wilcoxon rank sum test with continuity correction | SAC-SAL vs SAC-NIC, 6–9 min | W | 87.5000 | 0.0056* | |
| | Wilcoxon rank sum test with continuity correction | SAC-SAL vs NIC-SAL, 6–9 min | W | 178.0000 | 1.0000* | |
| | Wilcoxon rank sum test with continuity correction | SAC-NIC vs NIC-NIC, 6–9 min | W | 223.5000 | 1.0000* | |
| **J - CPP to 0.2 mg/kg nicotine - adult pretreatment** | | | | | | |
| SAC = 12, NIC = 12 | Shapiro-Wilk normality test | | W | 0.9361 | 0.1337 | Fig. 1I |
| | One Sample t test | SAC difference from 0 | t | 0.8596 | 0.4084 | |
| | One Sample t test | NIC difference from 0 | t | 0.7025 | 0.4969 | |
| | Welch Two Sample t-test | SAC vs NIC | t | 0.0065 | 0.9949 | |

*Holm correction for multiple comparisons.

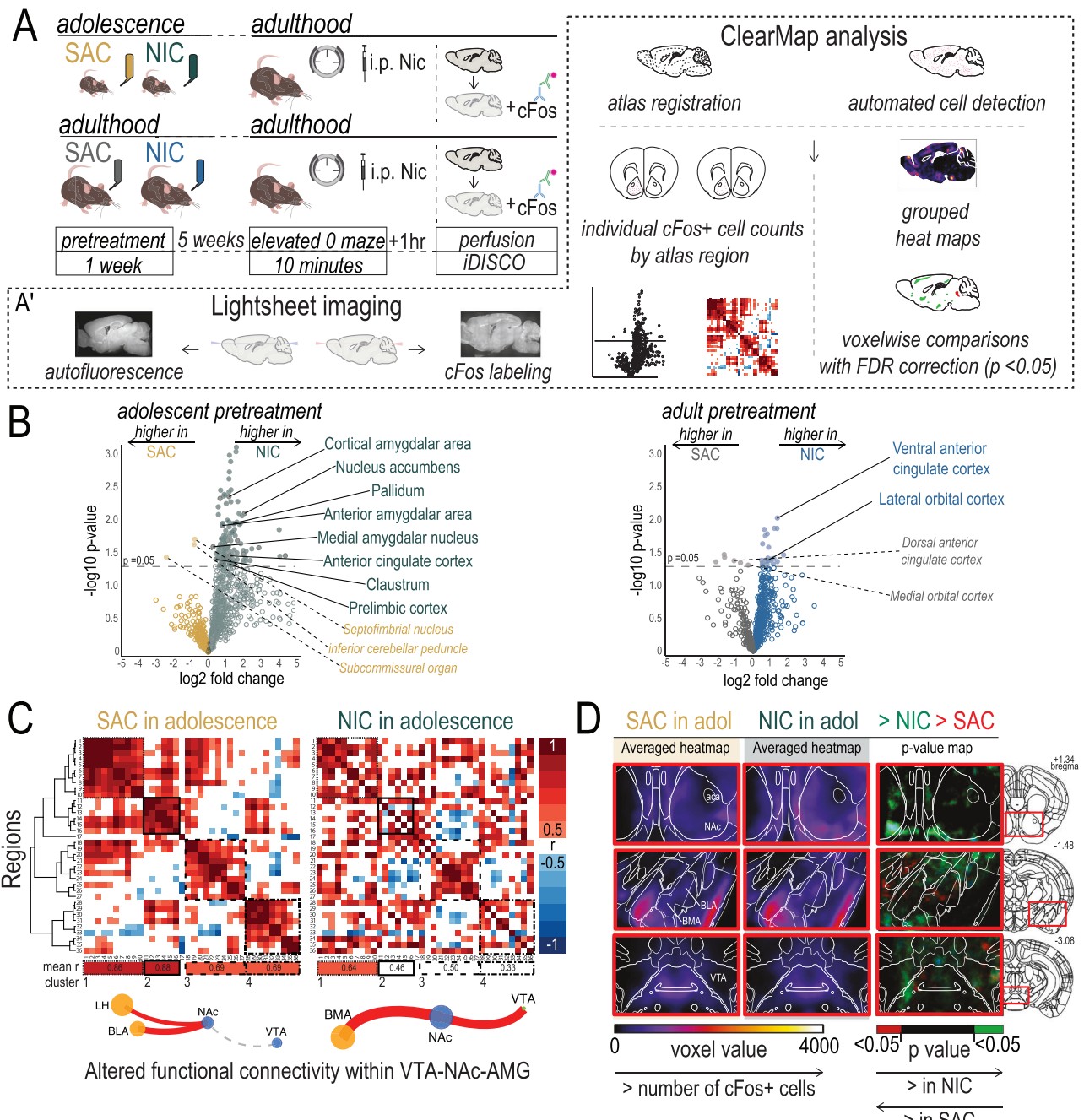

**Fig. 2 | *Nicotine in adolescence, but not in adulthood, reshapes brain-wide responsiveness to acute nicotine.* A** Experimental timeline for mice treated with Nicotine (NIC; 100 μg/mL nicotine in 2% saccharin) or Saccharin (SAC; 2% saccharin only) in early adolescence or in adulthood. **A'** iDISCO brain clearing and Clearmap activity mapping pipeline. **B** (*Left*) Mice treated with NIC in adolescence show greater cFos activity across the brain, and in specific brain regions, following an acute nicotine injection when compared to their SAC-treated counterparts. Overall, 129 regions showed an increase in cFos activity in NIC-pretreated mice, defined as a fold change >0 and a log $p$ > 1.3 (equivalent to $p < 0.05$). Notable regions of interest associated with anxiety response and/or response to nicotine have been highlighted. (*Right*) The activation profile of mice treated with NIC in adulthood is not substantially different from their SAC-treated counterparts. Cell counts were compared with independent two-sample Student's $t$ tests assuming unequal

variances, and $p$-values were converted to $q$-values to control for false discovery rate. **C** Correlation matrices of relationships between cFos cell numbers in response to nicotine injection. Activation across brain regions in SAC-pretreated mice (*left*) is highly correlated, and hierarchical clustering organizes these regions into 4 distinct modules with the strongest inter-region relationships. In NIC-pretreated mice (*right*), correlations between regions are less strong. *Bottom*, community analysis on networks formed from these correlation matrices indicate significant reorganization of the VTA-NAc-AMG connections. **D** Voxel-by-voxel analysis of these regions reveals increases in nicotine reactivity within the VTA and DA terminal regions. P-Value maps of significant differences between groups were generated from comparisons by independent two-sample Student's $t$ tests assuming unequal variances. Green or red voxels indicate a $p$ <0.05 after FDR correction. All statistical comparisons were two-sided. Source data are provided as a Source Data file.

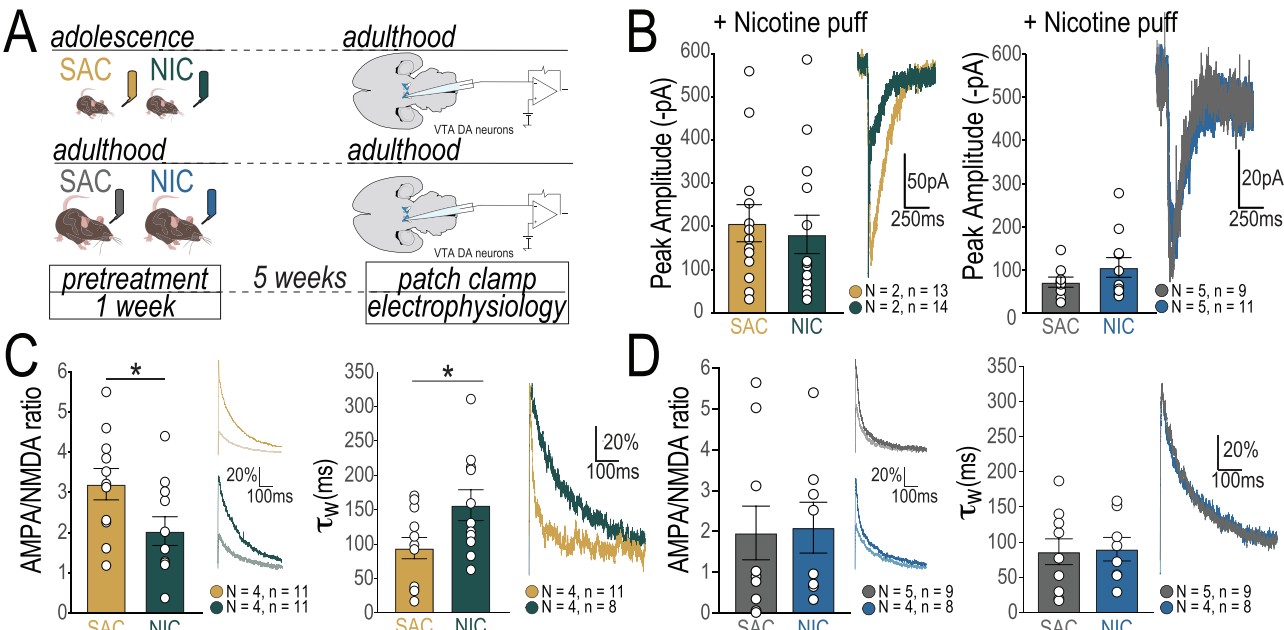

**Fig. 3 | Persistent immature neurophysiological signature of VTA dopamine neurons in mice exposed to NIC in adolescence. A** Experimental design for patch clamp experiments. **B** *Left*: No difference in peak amplitude of nicotine current between adult mice that received NIC or SAC as adolescents (NIC $N = 2$ mice, $n = 14$ neurons; SAC $N = 2$ mice, $n = 13$ neurons, Table 2A). *Right*: No difference in peak amplitude of nicotine current between adult mice that received NIC or SAC as adults (NIC $N = 5$ mice, $n = 11$ neurons; SAC $N = 5$ mice, $n = 9$ neurons, Table 2B). **C** *Left*: AMPA/NMDA ratio was decreased in mice that received NIC as adolescents (NIC $N = 4$ mice, $n = 11$ neurons; SAC $N = 4$ mice, $n = 11$ neurons, Table 2C). *Right*: NMDA current decay (weighted τ) was increased in mice that received NIC as

adolescents (NIC $N = 4$ mice, $n = 11$ neurons; SAC $N = 4$ mice, $n = 11$ neurons, Table 2D). **D** *Left*: No differences in AMPA/NMDA ratio were observed when mice received NIC as adults (NIC $N = 4$ mice, $n = 8$ neurons; SAC $N = 5$ mice, $n = 9$ neurons, Table 2E). *Right*: NMDA current decay (weighted τ) was the same between mice that received NIC or SAC as adults (NIC $N = 4$ mice, $n = 8$ neurons; SAC $N = 5$ mice, $n = 9$ neurons, Table 2F). All bar graphs are presented as mean values ± SEM. *$p < 0.05$, **$p < 0.01$, ***$p < 0.01$. All statistical comparisons were two-sided. Detailed information about statistical testing is available in Table 2. Source data are provided as a Source Data file.

received SAC or NIC as adults did not show differences in the voxel-by-voxel comparison of grouped cell counts across these same regions (Supplementary Fig. 2F). These results identify functional changes, particularly within dopaminoceptive networks, present in adult animals following exposure to NIC in adolescence, which may underlie the vulnerability to nicotine use we observed in these mice.

**Immature dopamine neuron physiology persists in adult mice exposed to nicotine in adolescence**

While we discovered significant perturbation in DAergic functional networks following adolescent exposure to NIC in our brain-wide activity analysis, including changes in activation in the NAc and AMG regions, we did not find changes in DA bouton density in these terminal regions (Supplementary Fig. 3A–D). Thus, we next assessed the electrophysiological profiles of DA neurons (Fig. 3A) to elucidate whether network changes may result from altered DA neuron responsivity to nicotine. The peak amplitude of nicotine currents did not differ between mice exposed to NIC or SAC in adolescence (Fig. 3B *left*, Table 2A) or adulthood (Fig. 3B *right*, Table 2B), however NIC treated mice showed faster response kinetics and an associated reduction in charge transferred (Supplementary Fig. 3E), suggesting that signaling through nAChR receptors may be altered in the VTA of these mice. Presynaptic release probability of excitatory synapses onto DA neurons did not appear to be altered in NIC exposed mice (Supplementary Fig. 3G), however we found striking differences in excitatory post-synaptic currents (Supplementary Fig. 3H) and plasticity on DA neurons only of adult mice exposed to NIC in adolescence. These mice showed a reduction in AMPA/NMDA ratio in comparison to SAC-treated controls (Fig. 3C *left*, Table 2C) and an increase in the weighted NMDA decay time constant (τ_w, Fig. 3C *right*, Table 2D), albeit without a clear contribution of changes to NR2B subunit expression

(Supplementary Fig. 3F). The changes were not present in mice exposed to nicotine as adults (Fig. 3D, Table 2E, F). Interestingly, these alterations to glutamatergic plasticity onto VTA dopamine neurons of adult mice exposed to NIC as adolescents resemble some conditions observed in young brains[16,41–44], suggesting that NIC exposure in adolescence may impede the maturation of glutamatergic plasticity mechanisms.

To determine if mice also maintain an adolescent-like response to nicotine injection after adolescent exposure, we first defined adult and adolescent responses to nicotine using in vivo single unit recordings in anesthetized mice (Fig. 4A). Dopamine neurons of naïve adult male mice show opposing responses to intravenous nicotine administration, with >90% of neurons projecting to the nucleus accumbens (NAc) activated in response to nicotine, and the >85% of neurons projecting to the amygdala (Amg) inhibited by nicotine in a previous study[23]. We first replicated this same distribution in adult mice, with DA neurons (identified by their electrophysiological characteristics and confirmed with neurobiotin labeling in a subset of neurons) showing both excitatory (putative VTA-NAc circuit) and inhibitory (putative VTA-Amg circuit) responses to i.v. nicotine injection (Fig. 4B, bottom). Next, we discovered that this pattern is already apparent in early adolescence, with DA neurons both activated or inhibited by nicotine present in the VTA (Fig. 4B, top). Dopamine neurons from naïve adolescent mice, however, showed a higher maximum activation in response to nicotine injection when compared to naïve adults (Fig. 4C, Table 3A), while there was no difference in the level of DA neuron inhibition by nicotine between the ages (Table 3B). The fact that DA neuron responses to nicotine in adolescent animals are "imbalanced", or are stronger in activation of the VTA-NAc pathway, may represent a physiological vulnerability signature, in line with reports of greater rewarding effects of nicotine in adolescent animals[8].

**Table 2 | Detailed Statistics for Fig. 3**

| N | Test | Factor | Statistic | Statistic Value | p value | Corresponding Figure |
|---|------|--------|-----------|-----------------|---------|----------------------|
| A - Nicotine current - adolescent pretreatment | | | | | | |
| NIC = 2 mice, 14 neurons; SAC = 2 mice, 13 neurons | Shapiro-Wilk normality test | | W | 0.8553 | 0.0015 | Fig. 3B |
| | Wilcoxon rank sum test with continuity correction | SAC vs NIC | W | 108.0000 | 0.4233 | |
| B - Nicotine current - adult pretreatment | | | | | | |
| NIC = 5 mice, 11 neurons; SAC = 5 mice, 9 neurons | Shapiro-Wilk normality test | | W | 0.8200 | 0.0017 | Fig. 3B |
| | Wilcoxon rank sum test with continuity correction | SAC vs NIC | W | 42.0000 | 0.5949 | |
| C - AMPA/NMDA Ratio - adolescent pretreatment | | | | | | |
| NIC = 4 mice, 11 neurons; SAC = 4 mice, 11 neurons | Shapiro-Wilk normality test | | W | 0.9603 | 0.4963 | Fig. 3C |
| | Welch Two Sample t-test | SAC vs NIC | t | −2.2008 | 0.0398 | |
| D - Weighted Tau - adolescent pretreatment | | | | | | |
| NIC = 4 mice, 8 neurons; SAC = 4 mice, 11 neurons | Shapiro-Wilk normality test | | W | 0.9474 | 0.2808 | Fig. 3C |
| | Welch Two Sample t-test | SAC vs NIC | t | 2.2914 | 0.0343 | |
| E - AMPA/NMDA Ratio - adult pretreatment | | | | | | |
| NIC = 4 mice, 8 neurons; SAC = 5 mice, 9 neurons | Shapiro-Wilk normality test | | W | 0.8319 | 0.0057 | Fig. 3D |
| | Wilcoxon rank sum test with continuity correction | SAC vs NIC | W | 39.5000 | 0.7727 | |
| F - Weighted Tau - adult pretreatment | | | | | | |
| NIC = 4 mice, 8 neurons; SAC = 5 mice, 9 neurons | Shapiro-Wilk normality test | | W | 0.9380 | 0.2949 | Fig. 3D |
| | Welch Two Sample t test | SAC vs NIC | t | 0.1470 | 0.8851 | |

We next assessed VTA dopamine neuron firing in adult mice exposed to SAC or NIC during adolescence or during adulthood (Fig. 4D). We found that all groups of mice showed a spectrum of response to nicotine as observed in naïve animals, with responses again ranging from strong activation to strong inhibition by nicotine (Fig. 4E). Interestingly, mice exposed to nicotine in adolescence showed a stronger, adolescent-like, activation of DA neurons in response nicotine than their SAC treated counterparts, with no change in the magnitude of inhibition (Fig. 4F, Table 3C, D). This result suggests that there is greater activation in the VTA-NAc pathway in the NIC pre-treated mice. This effect was not observed in mice exposed to nicotine as adults (Fig. 4G, Table 3E, F), suggesting that nicotine in adolescence promotes the persistence of an endogenous adolescent state when the effects of nicotine are skewed in favor of its rewarding effect.

## Restoring adult-like balance in nicotine-evoked DA signaling unmasks mature behavioral response

Our electrophysiological results raise the intriguing hypothesis that exposure to nicotine in adolescence "freezes" dopamine circuitry in an imbalanced, immature state, which promotes vulnerability to nicotine use. More concretely, this means that exposure to nicotine in adolescence prevented the maturation of the dopamine system, keeping its response to nicotine in a persistent adolescent-like state. We next sought to test whether adolescent-like DA functioning would explain the differences that we observed in the behavioral response to nicotine following preexposure in adolescence. We thus tested naïve adult and adolescent mice for CPP to a 0.2 mg/kg dose of nicotine and for nicotine-induced anxiety-like behavior in the EOM to establish whether there is a basal difference between adolescents and adults in these nicotine responses. Naïve adolescent or adult mice underwent a 5-day CPP paradigm where either one compartment was paired with saline and one with 0.2 mg/kg of nicotine, or both chambers were paired with saline. Multiple previous studies have shown that adolescent rodents show CPP to lower doses of nicotine than adult rodents do[45–47]. We

indeed found that adolescent mice showed a robust place preference to nicotine (Fig. 5A *center*, Table 4A), while adult mice showed no preference for the nicotine-paired chamber (Fig. 5A *right*, Table 4B), in line with our previous findings[32].

In accordance with our previous results[23], naïve adult mice showed a decrease in time spent in the open arms of the EOM following an i.p. injection of nicotine, but not after an injection of saline, indicating an anxiety-like response to nicotine (Fig. 5B *right*, Table 4D). However, when mice were tested at PND 28 this anxiety-like response to nicotine was absent, as the mice showed no difference in time spent in the open arms after nicotine or saline injection (Fig. 5B *center*, Table 4C), in line with results from adolescent rats[48]. As this strongly resembled the behavioral effect that we observed in mice treated with NIC in adolescence (Fig. 1E), this result suggests that not only does nicotine in adolescence block both neural and behavioral responses to nicotine in an immature state, but that imbalance between nicotine-induced activation and inhibition of VTA dopamine neurons could be actively masking the anxiogenic effect of the drug.

While increased place preference to nicotine in adolescent and adolescent-pretreated animals can be explained by the augmented nicotine response in NAc-projecting VTA dopamine neurons, as specifically stimulating this pathway promotes place preference[23], the mechanism underlying the abolishment of nicotine induced anxiety is unknown. If the exaggerated, adolescent-like VTA-NAc DA neuron response to nicotine in adult mice exposed to NIC as adolescents is masking the anxiogenic effects of the drug, dampening the activity of this circuit during nicotine administration could reveal the mature response. To test this possibility, we used a chemogenetic approach to exclusively decrease the excitability of VTA DA neurons of mice exposed to NIC in adolescence[49], by expressing an inhibitory DREADD (hM4D($G_i$)) in a projection- and neurotransmitter-specific manner (Fig. 5C). Mice that received the control virus recapitulated our previous results (Fig. 1E) of an abolished anxiogenic response to nicotine in adults previously exposed to NIC in adolescence (Fig. 5D *center*). In contrast, dampening VTA-NAc DA activity with the DREADD virus

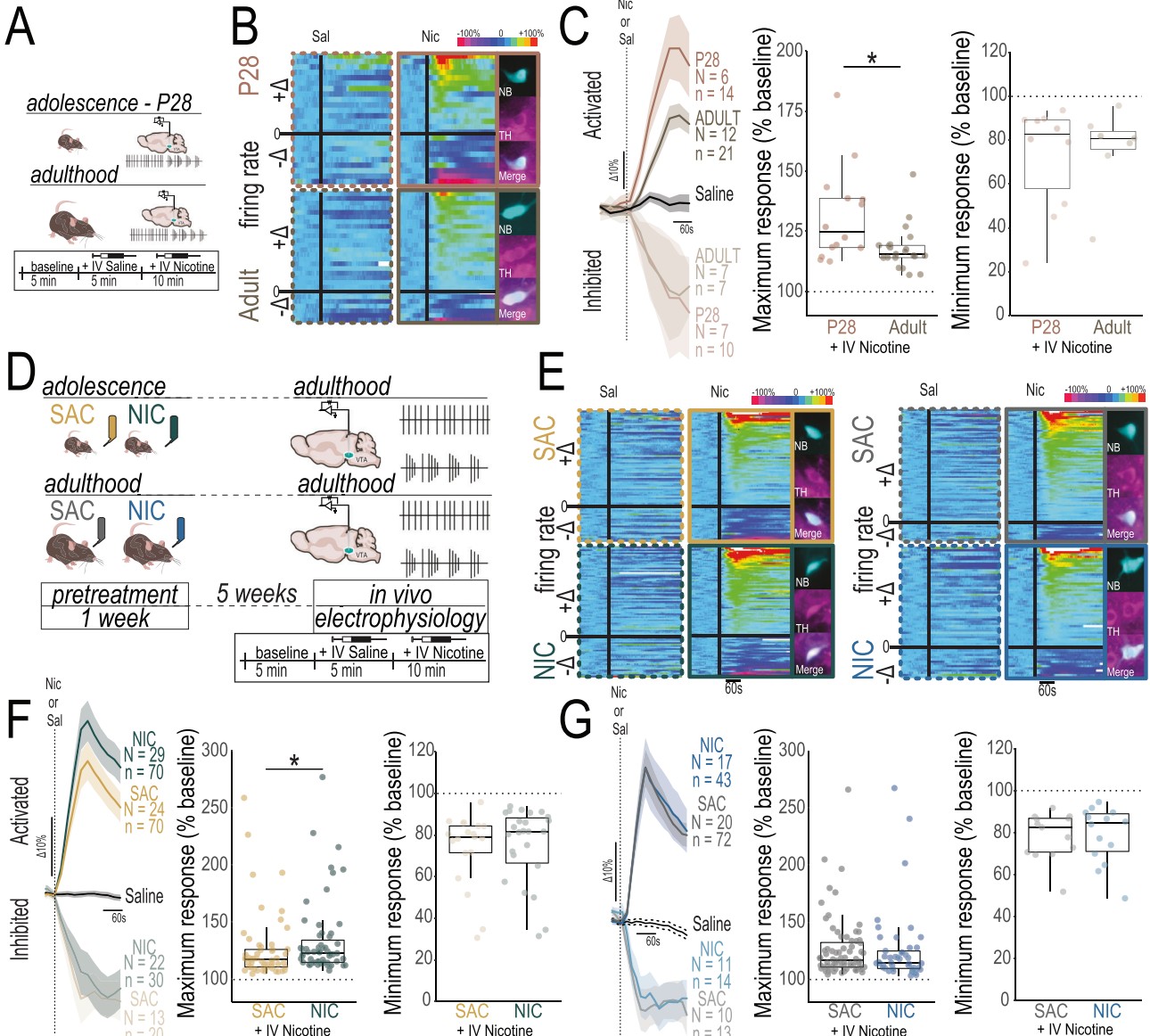

**Fig. 4 | An adolescent-like imbalance in VTA dopamine neuron response to nicotine persists in adult mice exposed to nicotine in adolescence.**
**A** Experimental design. **B** Neuron responses to saline (Sal) or nicotine (Nic) injection represented as changes from baseline activity in adolescent (P28) mice (*Top*) and adult (>P60) mice (*Bottom*). *Insets*: Example neurons were labeled with neurobiotin (NB) and tyrosine hydroxylase (TH) to confirm their dopaminergic identity. **C** Nicotine evoked activation was increased in dopamine neurons of adolescent mice in comparison with adults (*center*, Adolescent *N* = 6 mice, *n* = 14 neurons; Adult *N* = 12 mice, *n* = 21 neurons, Table 3A), with no difference in inhibition (*right*, Adolescent *N* = 7 mice, *n* = 7 neurons; Adult *N* = 7 mice, *n* = 10 neurons, Table 3B). **D** Experimental design. **E** Neuron responses to Sal or Nic injection in adolescent-treated mice (*left*) and adult-treated mice (*right*). *Insets*: NB + /TH+ example neurons. **F** Nicotine evoked activation was increased in dopamine neurons of adult

mice treated with NIC in adolescence in comparison with SAC-treated counterparts (*center*, NIC *N* = 29 mice, *n* = 70 neurons; SAC *N* = 24 mice, *n* = 70 neurons, Table 3C), with no difference in inhibition (*right*, NIC *N* = 22 mice, *n* = 30 neurons; SAC *N* = 13 mice, *n* = 20 neurons, Table 3D). **G** Nicotine evoked activation did not differ between dopamine neurons of mice treated with NIC or SAC as adults (*center*, NIC *N* = 17 mice, *n* = 43 neurons; SAC *N* = 20 mice, *n* = 72 neurons, Table 3E) nor in inhibition (*right*, NIC *N* = 11 mice, *n* = 14 neurons; SAC *N* = 10 mice, *n* = 13 neurons, Table 3F). Line graphs are presented with lines as mean values and shaded regions as ± SEM. Box plots include a box extending from the 25th to 75th percentiles, with the median indicated by a line and with whiskers extending from the minima to the maxima. *$p < 0.05$, **$p < 0.01$, ***$p < 0.01$. All statistical comparisons were two-sided. Detailed information about statistical testing is available in Table 3. Source data are provided as a Source Data file.

caused the mice to spend dramatically less time in the open arms of the EOM after i.p. nicotine injection (Fig. 5D *right*, Table 4E), an effect closely resembling the response to nicotine in naïve adults (Fig. 5B right) and in adult mice treated with SAC in adolescence (Fig. 1E). However, both mCherry and DREADD injected mice that received CNO one hour before being tested with saline in the EOM showed similar levels of exploration of the open arms, in line with our other experiments, indicating that (1) exposure to the CNO *itself* does not produce an anxiogenic or anxiolytic effect, and (2) decreasing the excitability of

the VTA-NAc pathway alone (i.e., the hM4D(Gi)-SAL condition) does not produce an anxiogenic or anxiolytic effect. The DREADD + nicotine mice were the only group tested that showed a significant anxiogenic response in the EOM, indicating that decreasing the excitability of VTA-NAc DA neurons restored the mature behavioral response to nicotine, effectively unmasking the anxiogenic effect of the drug. Together, our results suggest that NIC in adolescence perpetuates a developmental signaling imbalance between discrete DA circuits in response to nicotine which masks the anxiogenic effect of the drug, leading to a

**Table 3 | Detailed Statistics for Fig. 4**

| N | Test | Factor | Statistic | Statistic Value | p value | Corresponding Figure |
|---|------|--------|-----------|-----------------|---------|----------------------|
| A - maximum activation - naïve animals | | | | | | |
| Adolescent = 14, Adult = 21 | Shapiro-Wilk normality test | | W | 0.7851 | 0.0000 | Fig. 4C |
| | Wilcoxon rank sum test with continuity correction | Adolescent vs. Adult | W | 215.0000 | 0.0230 | |
| B - maximum inhibition - naïve animals | | | | | | |
| Adolescent = 10, Adult = 7 | Shapiro-Wilk normality test | | W | 0.8089 | 0.0027 | Fig. 4C |
| | Wilcoxon rank sum test with continuity correction | Adolescent vs. Adult | W | 37.0000 | 0.8836 | |
| C - maximum activation - adolescent pretreatment | | | | | | |
| NIC = 70, SAC = 68 | Shapiro-Wilk normality test | | W | 0.6534 | 0.0000 | Fig. 4F |
| | Wilcoxon rank sum test with continuity correction | NIC vs SAC | W | 1748.0000 | 0.0072 | |
| D - maximum inhibition - adolescent pretreatment | | | | | | |
| NIC = 30, SAC = 21 | Shapiro-Wilk normality test | | W | 0.7946 | 0.0000 | Fig. 4F |
| | Wilcoxon rank sum test with continuity correction | NIC vs SAC | W | 262.0000 | 0.3150 | |
| E - maximum activation - adult pretreatment | | | | | | |
| NIC = 45, SAC = 72 | Shapiro-Wilk normality test | | W | 0.6465 | 0.0000 | Fig. 4G |
| | Wilcoxon rank sum test with continuity correction | NIC vs SAC | W | 1810.0000 | 0.2884 | |
| F - maximum inhibition - adult pretreatment | | | | | | |
| NIC = 14, SAC = 13 | Shapiro-Wilk normality test | | W | 0.9032 | 0.0158 | Fig. 4G |
| | Wilcoxon rank sum test with continuity correction | NIC vs SAC | W | 84.0000 | 0.7524 | |

vulnerability to increased nicotine taking in adult animals. Furthermore, our findings from the DREADD experiments indicate that this imbalance can be rescued by targeted pathway manipulations.

## Discussion

Nicotine use in adolescence remains a serious public health issue, with strong associations between early onset of use and later addiction[4]. Moreover, the widespread availability and acceptability of e-cigarette use has facilitated a startling increase in number of adolescent users and a concomitant decrease in their age of onset[26,50]. Understanding how nicotine in adolescence impacts developing neural circuits is essential for informing evidence-based intervention efforts. We propose that exposure to nicotine in adolescence prolongs a developmental imbalance in dopaminergic circuitry and in behavioral response to nicotine re-exposure, which, in turn, can promote vulnerability to nicotine use and addiction.

Mesocorticolimbic DA circuits are known to undergo extraordinary growth and remodeling in the adolescent brain[11], in contrast to other neuromodulatory systems which are largely established at the start of adolescence. As such, the DA system is known to be sensitive to experience in adolescence, with significant effects of stress, social isolation, and drug use persisting until adulthood in many cases[10,11]. While VTA DA neurons are often considered as a single population, DA neurons in fact show significant physiological and molecular heterogeneity, which are associated with differences in their input-output wiring[43,51]. Increasingly, these diverse DA pathways are shown to play divergent roles in behavior, including responses to drugs of abuse[20,23,52,53]. With regards to nicotine response, we have shown that VTA-NAc projecting neurons are activated by nicotine, while VTA-Amg projecting neurons are inhibited by nicotine in adult mice, leading, respectively, to the expression of the reinforcing and anxiogenic properties of behavioral response to the drug[23]. How exactly adolescent experiences shape dopaminergic function within these distinct circuits is still an open question, with possibilities including to advance, slow, or misroute normal maturational processes. Here, our evidence suggests that experience with nicotine in adolescence arrests the development of VTA-NAc and VTA-AMg DA circuits, keeping them in a prolonged adolescent state. This stalling of developing DA circuits in response to experience is likely not limited to nicotine, or even drug exposure, as similar changes in glutamate plasticity onto VTA DA

neurons have been reported after food insecurity in adolescence[54]. This raises important questions in the cross-sensitivity and additive relationship between stress and drug use in adolescence and adulthood. Other studies have reported pathological miswiring of DA circuits after psychostimulant exposure in adolescence[10,25]. While our evidence does not support the idea that nicotine in early adolescence leads to the global miswiring of dopamine axons, as the innervation patterns of dopamine axons in the NAc and AMG are unchanged, we find, rather, the persistent maintenance of adolescent-like responses to an acute nicotine challenge. Importantly, we also provide evidence that adult-like behavior can be restored by artificially re-calibrating neural responses to adult levels.

Our 1-week nicotine exposure paradigm is not sufficient to cause behavioral or physiological changes when this exposure occurs during adulthood. However, when this same exposure regimen is administered during adolescence, it produces enduring alterations to the response to a later nicotine challenge, priming the VTA-NAc dopamine pathway, specifically, to be more reactive to nicotine. This is in line with work showing that the acute and enduring effects of nicotine on the adolescent brain differ from those in adults. Notably, adolescent rodents are thought to be more sensitive to the rewarding effects of nicotine, as well as less sensitive to its aversive effects, than adult animals[8,48]. The naturally occurring development of brain circuitry between adolescence and adulthood can thus be viewed as a protective mechanism against the deleterious effects of experience, and of nicotine use in particular. We found that the behavioral and physiological response to acute nicotine in adult mice pretreated with nicotine in adolescence strongly resembles the response of naïve adolescent mice. This finding suggests that nicotine exposure in adolescence blocks the protective normative maturation of nicotine-responsive systems and results in discordance between rewarding and aversive nicotine signals. We propose that this may be one mechanism by which nicotine use in adolescence creates an enduring vulnerability to later nicotine use and addiction. An imbalance between the rewarding and aversive effects of nicotine has indeed been posited as a mechanism driving the transition from casual use to addiction[29,55–57].

We focused our nicotine treatment on the early adolescent period (~PND21-28) because not only does this correspond to the time of life when human adolescents are most likely to begin nicotine use[26,27], but previous work in rodent models has also suggested that this is a

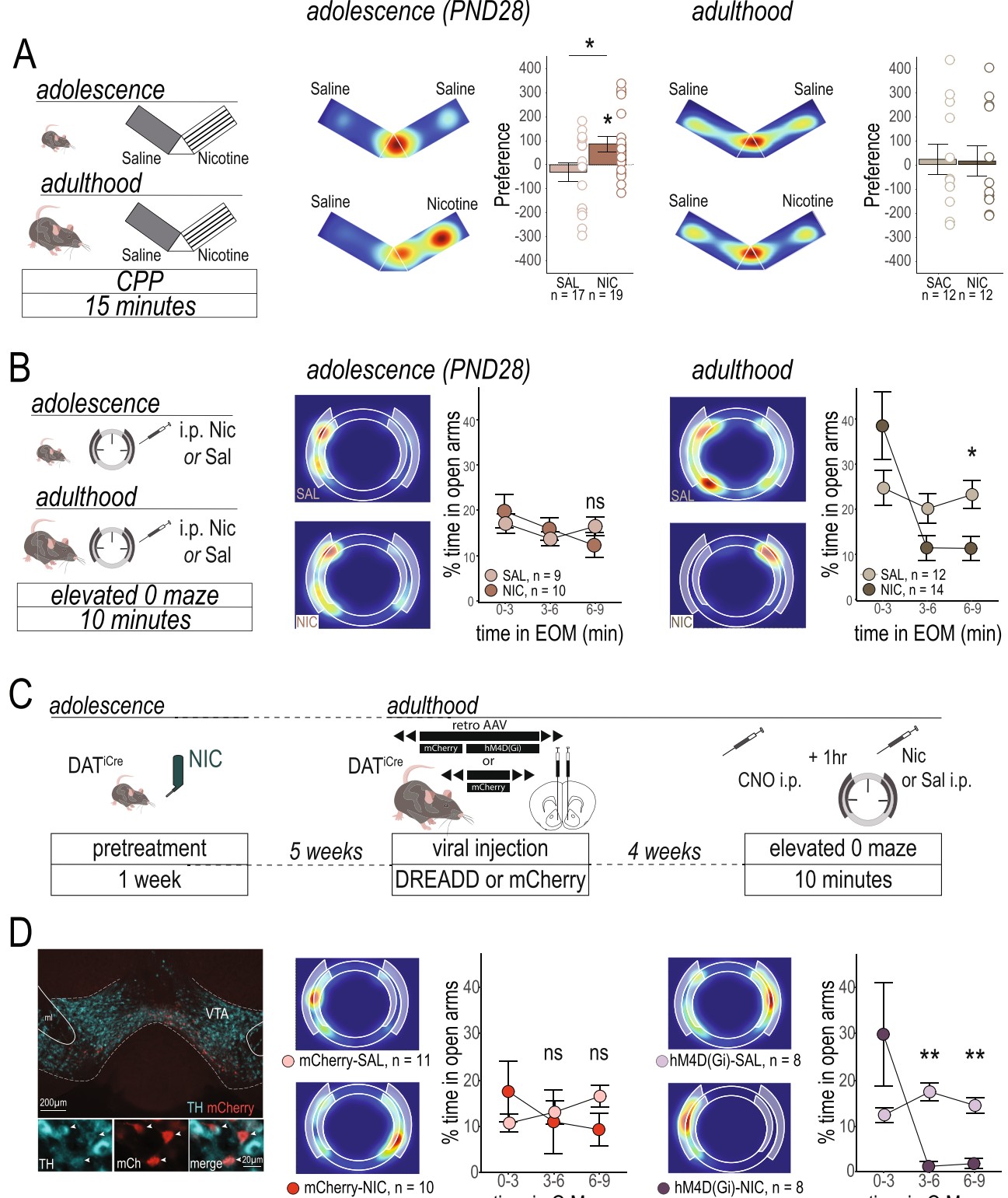

particularly vulnerable time for DA development,[25,58]. Importantly, marked sex differences exist in adolescent developmental trajectories, including dopamine development[11]. Recent work has shown that adolescent periods of DA circuit vulnerability to experience differ between male and female mice, and that even when the immediate effects of an experience are the same, the enduring outcomes may differ dramatically due to sex- and/or age specific compensatory processes[25]. Nicotine receptor expression and function can be modulated by female sex

hormones[59], suggesting that even the immediate experience or effects of nicotine may be sex-dependent, adding to the complexity of addressing sex and developmental interactions in its effects. Ongoing work will address whether female mice show a similar or different pattern of adolescent DA circuit "freezing" by nicotine. Differential ages of vulnerability between sexes are also a key research arena, taking into account data from human studies showing that boys begin using nicotine younger than girls do[60], and are more likely to

**Fig. 5 | Chemogenetically dampening VTA-NAc circuit activity restores anxiety-like response to nicotine in adult mice exposed to nicotine in adolescence.**
**A** Experimental timeline for naïve mice (*left*). Following a 5-day CPP paradigm, adolescent mice showed a significant place preference for the chamber paired with 0.2 mg/kg nicotine (*center*, Table 4A), while adult mice did not (*right*, Table 4B). **B** Experimental timeline for naïve mice (*left*). Adolescent mice show no difference in the time spent in the open arms of the EOM between a saline or nicotine injection (*center*, Table 4C). Adult mice spend less time in the open arms of the EOM following an injection of nicotine than an injection of saline (*right*, Table 4D), indicative of an anxiogenic effect of nicotine. **C** Experimental timeline for DREADD intervention experiment. DAT[iCre] mice were pretreated with NIC in adolescence, and then as adults they received an injection of a retroAAV hM4D(Gi) or control fluorescent-reporter virus into the NAc at the level of the medial shell. After 4 weeks, mice received an injection of CNO one hour before an injection of nicotine

or saline and entering the EOM. **D** Retro AAV viruses were well expressed in VTA DA neurons of DAT[iCre] mice following their behavioral testing (*left*). Mice that received a control virus replicated the effect of NIC in adolescence on WT mice, as mice never spent less time in the open arms of the EOM than their saline-treated counterparts (*center*, Table 4E). When VTA-NAc DA activity was reduced before nicotine injection, however, a mature behavioral response to nicotine injection, where mice spend less time in the open arms of the EOM, was restored (*right*, Table 4E). All line graphs are presented as mean values ± SEM. Graphs are separated by pre-treatment group for clarity, but statistical analyses compared all four treatment conditions. Heatmaps are from representative individual animals. *$p < 0.05$, **$p < 0.01$, ***$p < 0.01$, ns = not significant. All statistical comparisons were two-sided. Holm's sequential Bonferroni corrections were used to correct for multiple comparisons. Detailed information about statistical testing is available in Table 4. Source data are provided as a Source Data file.

escalate nicotine intake by transitioning from e-cigarette to cigarette use[61].

Why adolescent nicotine users are more likely to continue to use nicotine, to have longer and heavier smoking careers, and to develop cross-drug addictions and/or psychiatric symptoms has largely remained unexplained at a mechanistic level. The oral nicotine consumption paradigm used in the current studies cannot be considered a direct model of adolescent nicotine use in humans, notably as the route of nicotine administration (and thus its pharmacokinetic properties[62]), the dosing, and the metabolism of the drug differ between species. However, here we are able to show that exposure to nicotine during a discrete period in adolescence locks the function of DA neurons in a persistent adolescent-like state, leading to exaggerated response to the rewarding effects of nicotine and a blunting of its negative, anxiogenic effects. This adolescent-like bias in behavioral responding may therefore facilitate the transition from casual use to addiction across the lifetime. Because we were able to restore an adult-like behavioral response to nicotine with a chemogenic approach, our findings suggest that these effects of nicotine in adolescence may be reversible. Developing interventions that target restoring an appropriate balance between the positive and negative effects of nicotine use may have therapeutic applications.

## Methods

### Animals
All experiments and procedures were performed in accordance with European Commission directives 219/1990, 220/1990 and 2010/63, and approved by Sorbonne Université and the ethical committee (CEEA) #005 under the protocol number 00553.02, or by the ESPCI and the ethical committee #059 under APAFIS#24037-2020102011276492. Wild-type (WT) C57BL/6 mice (Janvier Labs, France) or DATiCre mice (from François Tronche[63]) were maintained on a 12-h light–dark cycle (light on at 0800 h), at an average temperature of 21 degrees and -50% humidity. Mice were given ad libitum access to food and water unless noted.

### Drugs
**Pretreatment regimen.** Nicotine tartrate salt (Glentham Biosciences) was dissolved into a 2% Saccharin (Sigma) solution to a final concentration of 100 μg/mL and pH was adjusted to 7.2 ± 0.2. Mice had access to either the nicotine solution or a 2% saccharin solution during one week.

**Experimental solutions.** Nicotine tartrate salt was solubilized in a physiological saline solution (0.9% NaCl) with pH adjusted to 7.2 ± 0.2, and was injected intravenously (IV) at a 30 μg/kg dose for juxtacellular recordings, or intra-peritoneally (IP) at a 0.5 mg/kg dose for the elevated O-maze (EOM) test and for the activity mapping experiment. For the Two bottle choice task, Nicotine tartrate salt was dissolved in water to a final concentration of 10 μg/mL, 50 μg/mL, 100 μg/mL, or

200 μg/mL and pH was adjusted to 7.2 ± 0.2. All concentrations are expressed as free base.

### Behavior
**Two-bottle choice.** Mice were single-housed in a cage with two drinking bottles where the volume change was measured continuously every minute with an automated acquisition system (TSE system, Germany). Mice were first presented with water in both bottles for a four-day habituation period, and the position of the bottles (i.e.,. right or left side of the cage) was swapped after 2 days. For each of the test solutions the same pattern was used; mice had access to each of the solutions tested during 4 consecutive days, and the position of the test solution was changed every 2 days (Fig. 1A). Mice were first tested for sucrose preference by changing the solution in one bottle to a 0.5% sucrose solution, while the other continued to contain plain water. After 4 days, the test solution was changed to 1% sucrose. Nicotine preference was tested with consecutively increasing doses of nicotine (10, 50, 100 and 200 μg/ml, free base) in the test bottle, the control bottle remained full of plain water. Sweetening the nicotine and control solution with saccharin was avoided as it can obscure the difference in value between the two choices[64–67]. Mice were weighed every other day to quantify the nicotine intake in mg/kg/day. Mice showing a strong side bias (preference <20% or >80% for one side) during the habituation period were removed from the analyses. While measurements took place continuously, sessions were defined as running overnight each test day from 20:00 to 14:00, encompassing the most active drinking periods of the day (Fig. 1B) and homogenizing the measurement periods between days with continuous recording and days when the recording had to be stopped in order to change the position and/or contents of the bottles. All solution changes thus occurred during the 14:00-20:00 'off' time window. Minute-by-minute measurements were thresholded at a value of 0.1 ml, with values greater than 0.1 ml/min representing less than 0.01% of the full data set and considered to be erroneous measurements (leaking, sensor issue).

**Elevated O-maze test.** All behavioral tests were conducted during the light period of the animal cycle (between 1:00 and 7:00PM). The raw data for behavioral experiments were acquired as video files. The elevated O-maze (EOM) apparatus consists of two open (stressful) and two enclosed (protecting) elevated arms that together form a zero or circle (diameter of 50 cm, height of 58 cm, 10 cm-wide circular platform). Time spent in exploring enclosed versus open arms indicates then the anxiety level of the animal. The test lasts 10 min: mice are injected 1 min before the test, and then put in the EOM for 9 min. Mice are placed in an open arm at one of the four entrances of a closed arm, with the entrance (e.g.,1–4) distributed pseudorandomly across the experiment. Time spent in open or closed arms was extracted frame-by-frame using the open-source video analysis pipeline ezTrack[68]. Mice were habituated to the stress of handling and injection for a minimum of one week before testing.

**Table 4 | Detailed Statistics for Fig. 5**

| N | Test | Factor | Statistic | Statistic Value | p value | Corresponding Figure |
|---|------|--------|-----------|-----------------|---------|----------------------|
| A - CPP to 0.2 mg/kg nicotine - naive adolescents | | | | | | |
| SAL = 17, NIC = 19 | Shapiro-Wilk normality test | | W | 0.9712 | 0.4605 | Fig. 4A center |
| | One Sample *t* test | SAL difference from 0 | t | −0.7875 | 0.4425 | |
| | One Sample *t* test | NIC difference from 0 | t | 2.7048 | 0.0145 | |
| | Welch Two Sample *t* test | SAC vs NIC | t | 2.3317 | 0.0262 | |
| B - CPP to 0.2 mg/kg nicotine - naive adults | | | | | | |
| SAC = 12, NIC = 12 | Shapiro-Wilk normality test | | W | 0.8885 | 0.0124 | Fig. 4A right |
| | One Sample t-test | SAC difference from 0 | t | 0.3367 | 0.7427 | |
| | One Sample t-test | NIC difference from 0 | t | 0.2286 | 0.8234 | |
| | Welch Two Sample t-test | SAC vs NIC | t | −0.0797 | 0.9372 | |
| C - % time in open arms of EOM - naïve adolescent animals | | | | | | |
| SAL = 9, NIC = 10, | Shapiro-Wilk normality test | | W | 0.9832 | 0.6108 | Fig. 4B center |
| | Welch Two Sample t-test | NIC vs. SAC in 0–3 min | t | 0.6447 | 0.9308* | |
| | Welch Two Sample t-test | NIC vs. SAC in 3–6 min | t | 0.7492 | 0.9308* | |
| | Welch Two Sample t-test | NIC vs. SAC in 6–9 min | t | −1.2011 | 0.7408* | |
| D - % time in open arms of EOM - naïve adult animals | | | | | | |
| SAL = 12, NIC = 14, | Shapiro-Wilk normality test | | W | 0.8876 | 0.0000 | Fig. 4B right |
| | Wilcoxon rank sum test with continuity correction | NIC vs. SAC in 0–3 min | W | 104.0000 | 0.3159* | |
| | Wilcoxon rank sum test with continuity correction | NIC vs. SAC in 3–6 min | W | 51.0000 | 0.1891* | |
| | Wilcoxon rank sum test with continuity correction | NIC vs. SAC in 6–9 min | W | 35.0000 | 0.0377* | |
| E - % time in open arms of EOM - DREADDS | | | | | | |
| MCH + SAL = 11, MCH + NIC = 10, | Shapiro-Wilk normality test | | W | 0.7376 | 0.0000 | Fig. 4C |
| HM4 + SAL = 11, HM4 + NIC = 14 | Kruskal-Wallis rank sum test | All groups, 0–3 min | Kruskal-Wallis chi-squared | 1.7264 | 0.6311 | |
| | Kruskal-Wallis rank sum test | All groups, 3–6 min | Kruskal-Wallis chi-squared | 18.0767 | 0.0004 | |
| | Kruskal-Wallis rank sum test | All groups, 6–9 min | Kruskal-Wallis chi-squared | 16.4805 | 0.0009 | |
| | Wilcoxon rank sum test with continuity correction | HM4-SAL vs HM4-NIC, 3–6 min | W | 1.0000 | 0.0033* | |
| | Wilcoxon rank sum test with continuity correction | MCH-SAL vs MCH-NIC, 3–6 min | W | 29.0000 | 0.1846* | |
| | Wilcoxon rank sum test with continuity correction | MCH-SAL vs HM4-SAL, 3–6 min | W | 56.0000 | 0.1846* | |
| | Wilcoxon rank sum test with continuity correction | MCH-NIC vs HM4-NIC, 3–6 min | W | 21.0000 | 0.3423* | |
| | Wilcoxon rank sum test with continuity correction | HM4-SAL vs HM4-NIC, 6–9 min | W | 1.0000 | 0.0040* | |
| | Wilcoxon rank sum test with continuity correction | MCH-SAL vs MCH-NIC, 6–9 min | W | 28.0000 | 0.1368* | |
| | Wilcoxon rank sum test with continuity correction | MCH-SAL vs HM4-SAL, 6–9 min | W | 37.0000 | 0.1368* | |
| | Wilcoxon rank sum test with continuity correction | MCH-NIC vs HM4-NIC, 6–9 min | W | 18.0000 | 0.5915* | |

*Holm correction for multiple comparisons.

**Conditioned place preference (CPP).** CPP experiments were performed in a Y-maze apparatus (Imetronic, Pessac, France) with one closed arm and two other arms with manually operated doors as previously[31,32]. Briefly, two rectangular chambers (11 × 25 cm) with different cues (texture and color) are separated by a center triangular compartment (side of 11 cm) used as a neutral compartment. One pairing compartment textured black and white striped walls and floor, while the other has smooth gray floor and walls. The CPP apparatus was illuminated at 100 lux during the experiments. The first day (pretest) of the experiment, without previous habituation to the apparatus, mice explored the environment for 900 s (15 min) and the time spent in each compartment was recorded. Conditioning was unbiased: pretest data were used to segregate the animals with equal bias so each group has an initial preference score almost null, indicating no preference on average. On day 2, 3 and 4, animals received an i.p. injection of saline (0.9%NaCl) and were immediately confined to one of the pairing chambers for 1200 s (20 min) in the morning. Groups were balanced so that animals did not all receive nicotine in the same chamber. On the afternoon of the same day mice received an injection of 0.2 mg/kg nicotine or saline and were placed in the

opposite pairing chamber. On day 5 (test), animals were allowed to freely explore the whole apparatus for 900 s (15 min). Pretest and test sessions were videorecorded and analyzed using ezTrack. The preference score is expressed in seconds and is calculated by subtracting pretest from test data. Mice were habituated to the stress of handling and injection for a minimum of one week before testing.

## Brain clearing and activity mapping

**Experimental design and perfusion.** Mice were injected with i.p. saline or nicotine and kept in a dim, quiet room for one hour before perfusion to minimize off-target cFos expression. Mice were then perfused with 1X PBS followed by 20 mL of 4% paraformaldehyde (PFA, Electron Microscopy Services). Brains were carefully dissected from the skull, and stored in PFA overnight. Brains were stored in PBS with 0.01% Sodium Azide (Sigma-Aldrich, Germany) until clearing.

**iDISCO+ whole brain immunolabeling.** Whole brain clearing and immunostaining was performed following the iDISCO+ protocol previously described previously[33] with minimal modifications. All the steps of the protocol were done at room temperature with gentle shaking unless otherwise specified. All the buffers were supplemented with 0.01% Sodium Azide (Sigma-Aldrich, Germany) to prevent bacterial and fungal growth. Briefly, perfused brains were dehydrated in an increasing series of methanol (Sigma-Aldrich, France) dilutions in water (washes of 1 h in methanol 20%, 40%, 60%, 80% and 100%). An additional wash of 2 h in methanol 100% was done to remove residual water. Once dehydrated, samples were incubated overnight in a solution containing a 66% dichloromethane (Sigma-Aldrich, Germany) in methanol, and then washed twice in methanol 100% (4 h each wash). Samples were then bleached overnight at 4 °C in methanol containing a 5% of hydrogen peroxide (Sigma-Aldrich). Rehydration was done by incubating the samples in methanol 60%, 40% and 20% (1 h each wash). After methanol pretreatment, samples were washed in PBS twice 15 min and 1 h in PBS containing a 0,2% of Triton X-100 (Sigma-Aldrich) and further permeabilized by a 24 h incubation at 37 °C in Permeabilization Solution, composed by 20% dimethyl sulfoxide (Sigma-Aldrich), 2,3% Glycine (Sigma-Aldrich, USA) in PBS-T. In order to start the immunostaining, samples were first blocked with 0,2% gelatin (Sigma-Aldrich) in PBS-T for 24 h at 37 °C, the same blocking buffer was used to prepare antibody solutions. Brains were incubated with anti c-Fos primary (Synaptic systems 226-003) for 10 days at 37 °C with gentle shaking, then washed in PBS-T (twice 1 h and then overnight), and finally newly incubated for 10 days with secondary antibodies. Secondary antibodies raised in donkeys, conjugated to Alexa 647 were used (Life Technologies). After immunostaining, the samples were washed in PBS-T (twice 1 h and then overnight), dehydrated in a methanol/water increasing concentration series (20%, 40%, 60%, 80%, 100% one hour each and then methanol 100% overnight), followed by a wash in 66% dichloromethane – 33% methanol for 3 h. Methanol was washed out with two final washes in dichloromethane 100% (15 min each) and finally the samples were cleared and stored in dibenzyl ether (Sigma-Aldrich) until light sheet imaging.

**Light sheet microscopy.** The acquisitions were done on a LaVision Ultramicroscope II equipped with infinity-corrected objectives. The microscope was installed on an active vibration filtration device, itself put on a marble compressed-air table. Imaging was done with the following filters: 595/40 for Alexa Fluor-555, and −680/30 for Alexa Fluor-647. The microscope was equipped with the following laser lines: OBIS-561nm 100 mW, OBIS-639nm 70 mW, and used the 2nd generation LaVision beam combiner. The images were acquired with an Andor CMOS sNEO camera. Main acquisitions were done with the LVMI-Fluor 4X/O.3 WD6 LaVision Biotec objective. The microscope was connected to a computer equipped with SSD drives to speed up

the acquisition. The brain was positioned in sagittal orientation, cortex side facing the light sheet, to maximize image quality and consistency. A field of view of 1000 × 1300 pixels was cropped at the center of the camera sensor. The light sheet numerical aperture was set to NA-0.03. The 3 light sheets facing the cortex were used, while the other side illumination was deactivated to improve the axial resolution. Beam width was set to the maximum. Laser powers were set to 40−60% (639 nm). The center of the light sheet in x was carefully calibrated to the center of the field. z steps were set to 6 mm. Tile overlaps were set to 10%. The whole acquisition takes about 1 h per hemisphere. At the end of the acquisition, the objective is changed to a MI PLAN 1.1X/0.1 for the reference scan at 488 nm excitation (tissue autofluorescence). The field of view is cropped to the size of the brain, and the z-steps are set to 6 mm, and light sheet numerical aperture to 0.03 NA. It is important to crop the field of view to the size of the brain for subsequent alignment steps.

**Computing resources.** The data were automatically transferred every day from the acquisition computer to a Lustre server for storage. The processing with ClearMap was done on local workstations, either Dell Precision T7920 or HP Z840. Each workstation was equipped with 2 Intel Xeon Gold 6128 3.4 G 6 C/12 T CPUs, 512 Gb of 2666 MHz DDR4 RAM, 4x1Tb NVMe Class 40 Solid State Drives in a RAID0 array (plus a separate system disk), and an NVIDIA Quadro P6000, 24 Gb VRAM video card. The workstations were operated by Linux Ubuntu 20.04LTS. ClearMap 2.0 was used on Anaconda Python 3.7 environment.

**ClearMap Fos+ cell counting.** Tiled acquisitions of Fos-immunolabeled iDISCO+ cleared brains scanned with the light sheet microscope were processed with ClearMap 2 to generate both voxel maps of Fos cell densities, as well as region-based statistics of cell counts[34,69]. Briefly, stitched images were processed for background removal, on which local maxima were detected to place initial seeds for the cells. A watershed was done on each seed to estimate the volume of the cell, and the cells were filtered according to their volume to exclude smaller artefactual maxima. The alignment of the brain to the Allen Brain Atlas (March 2017) was based on the acquired autofluorescence image using Elastix (https://elastix.lumc.nl). Filtered cell's coordinates were transformed to their reference coordinate in the Allen Brain Atlas common coordinate system[70]. For voxel maps, spheres of 375 mm diameter were drawn on each filtered cell. P Value maps of significant differences between groups were generated using Mann−Whitney U test (SciPy implementation). Aligned voxelized datasets from each group of animals were manually inspected to identify the regional overlaps of p-value clusters, and volcano-plots of regional counts where generated.

## Quantitative neuroanatomy of dopamine terminals

**Tissue preparation.** Mice were perfused with 1X PBS followed by 20 mL of 4% paraformaldehyde (Sigma). Brains were postfixed in PFA overnight, transferred to PBS before sectioning. Serial 50 μm thick coronal sections were made on a Leica vibratome (VTS1000) within one week of perfusion, and stored free-floating in PBS. Free-floating brain sections were then incubated for 1 h at 4 °C in a blocking solution of phosphate-buffered saline (PBS) containing 3% bovine serum albumin (BSA, Sigma; A4503) (vol/vol) and 0.2% Triton X-100 (vol/vol), and then incubated overnight at 4 °C with a sheep anti-tyrosine hydroxylase antibody (anti-TH, Milipore, Ab1542) diluted 1:500 in PBS containing 1.5% BSA and 0.2% Triton X-100. The following day, sections were rinsed with PBS, and then incubated for 3 h at room temperature with Cy3-conjugated anti-sheep secondary antibody (Jackson ImmunoResearch, 713-165-147) diluted 1:500 in a solution of 1.5% BSA in PBS. After three rinses in PBS, slices were wet-mounted using Prolong Gold Antifade Reagent (Invitrogen, P36930).

**Confocal image acquisition and deconvolution.** Images stacks were taken with a Confocal Laser Scanning Microscope (A1, Nikon) equipped with a 60×1.4 NA objective (oil immersion, Nikon) with pinhole aperture set to 1 Airy Unit, pixel size of 50 nm and z-step of 200 nm. Excitation wavelength and emission range for Cy-3 labeling was: ex. 561, em. 570- 630. Laser intensity was set so that each image occupies the full dynamic range of the detector. Deconvolution using Maximum Likelihood Estimation algorithm was performed with Huygens software (Scientific Volume Imaging)[71]. 150 iterations were applied in classical mode, background intensity was averaged from the voxels with lowest intensity, and signal to noise ratio values were set to a value of 20.

**Segmentation of presynaptic boutons from confocal images.** Segmentation of TH+ boutons was performed in 3D in FIJI (fiji.net) using a procedure based on image analysis tools developed in the ImageJ plugin 3DImageSuite[72–74]. First local maxima were detected. Histogram analysis and image inspection allowed the definition of a threshold intensity so that only local maxima from presynaptic objects were retained. The local maxima were then used as seeds around which 3D intensity distribution was fitted to a Gaussian curve. The intensity value which defines 95% of the area of the gauss curve was chosen as threshold for the border of the object. Buttons of similar size but different intensity are thus extracted as object of similar size. The border of the object was determined following the so-called block algorithm; starting around the local maxima, voxels which followed three criteria were included in the object (intensity above the threshold, intensity lower than previously included voxel, and the inclusion is validated if neighboring voxels are included as well). Counts were used to determine a density for the obtained image stack and normalized to a standard volume of $10\,\mu m \times 10\,\mu m \times 10\,\mu m$ to make comparisons between images with different stack sizes, then exported to.csv.

**In vivo electrophysiology**
Mice were deeply anaesthetized with (1) an IP injection of chloral hydrate (400 mg/kg), supplemented as required to maintain optimal anesthesia throughout the experiment, or (2) isoflurane delivered continuously (5% induction, 2−3% maintenance; TEMSega). The scalp was opened and a hole was drilled in the skull above the location of the VTA. Intravenous administration of saline or nicotine (30 μg/kg) was carried out through a catheter (30 G needle connected to polyethylene tubing PE10) connected to a Hamilton syringe, into the saphenous vein of the animal. Extracellular recording electrodes were constructed from 1.5 mm outer diameter/1.17 mm inner diameter borosilicate glass tubing (Harvard Apparatus) using a vertical electrode puller (Narishige). The tip was broken straight and clean under microscopic control to obtain a diameter of about 1 μm. The electrodes were filled with a 0.5% NaCl solution containing 1.5% of neurobiotin® tracer (VECTOR laboratories) yielding impedances of 6−9 MΩ. Electrical signals were amplified by a high-impedance amplifier (Axon Instruments) and monitored audibly through an audio monitor (A.M. Systems Inc.). The signal was digitized, sampled at 25 kHz, and recorded on a computer using Spike2 software (Cambridge Electronic Design) for later analysis. The electrophysiological activity was sampled in the central region of the VTA (coordinates: between 3.1 and 4 mm posterior to bregma, 0.3 to 0.7 mm lateral to midline, and 4 to 4.8 mm below brain surface). Individual electrode tracks were separated from one another by at least 0.1 mm in the horizontal plane. Spontaneously active DA neurons were identified based on previously established electrophysiological criteria[75].

After recording, a subset of nicotine-responsive cells were labeled by electroporation of their membrane: successive currents squares were applied until the membrane breakage, to fill the cell soma with neurobiotin contained into the glass pipet[76]. To be able to establish correspondence between neurons responses and their localization in the VTA, we labeled one type of response per mouse: solely activated neurons or solely inhibited neurons, with a limited number of cells per brain (1 to 4 neurons maximum, 2 by hemisphere), always with the same concern of localization of neurons in the VTA.

**Ex vivo patch-clamp recordings**
Mice were anesthetized (Ketamine 150 mg/kg; Xylazine 10 mg/kg) and transcardially perfused with aCSF for slice preparation. For VTA recordings, horizontal 250 μm slices were obtained in bubbled ice-cold 95% $O_2$/5% $CO_2$ aCSF containing (in mM): KCl 2.5, NaH2PO4 1.25, MgSO4 10, CaCl2 0.5, glucose 11, sucrose 234, NaHCO$_3$ 26. Slices were then incubated in aCSF containing (in mM): NaCl 119, KCl 2.5, NaH$_2$PO$_4$ 1.25, MgSO$_4$ 1.3, CaCl$_2$ 2.5, NaHCO$_3$ 26, glucose 11, at 37 °C for 1 h, and then kept at room temperature. Slices were transferred and kept at 32–34 °C in a recording chamber superfused with 2.5 ml/min aCSF. Visualized whole-cell voltage-clamp recording technique was used to measure synaptic responses using an upright microscope (Olympus France). Putative DA neurons were recorded in the lateral VTA and identified using criteria such as localization and cell body size, as well as electrophysiological signature (e.g., broad action potential, and large Ih current)[75]. Spontaneous excitatory postsynaptic currents (sEPSCs) induced by puffing nicotine onto DA cells were measured, as well as analyses of paired-pulse ratios (PPR) and AMPA-R/NMDA-R ratios as an index of synaptic adaptations.

Experiments were obtained using a Multiclamp 700B (Molecular Devices, Sunnyvale, CA). Signals were collected and stored using a Digidata 1440 A converter and pCLAMP 10.2 software (Molecular Devices, CA). In all cases, analyses were performed using Clampfit 10.2 (Axon Instruments, USA) and Prism (Graphpad, USA).

Paired-pulse ratios (PPR) and AMPA-R/NMDA-R ratios were assessed in voltage-clamp mode using an internal solution containing (in mM) 130 CsCl, 4 NaCl, 2 MgCl$_2$, 1.1 EGTA, 5 HEPES, 2 Na2ATP, 5 sodium creatine phosphate, 0.6 Na3GTP, and 0.1 spermine. The PPR protocol consisted in two evoked pulses 50 ms apart applied every 15 s at V = −60 mV. Synaptic currents were evoked by stimuli (10 μs) at 0.15 Hz through a glass pipette placed 200 μm from the patched neurons. Voltage was then raised to V = +40 mV and synaptic currents were evoked in the absence and in the presence of AMPA-R antagonist DNQX as previously described for AMPA-R/NMDA-R ratios[77].

To assess NMDA Currents, voltage-clamp were performed using an internal solution containing (in mM) 130 CsCl, 4 NaCl, 2 MgCl$_2$, 1.1 EGTA, 5 HEPES, 2 Na2ATP, 5 sodium creatine phosphate, 0.6 Na3GTP, and 0.1 spermine. Cells were held at +40 mV and NMDA synaptic currents were evoked 15 times (3 times per min) by stimuli of 0.1 ms every 20 s through a glass pipette placed 200 μm from the patched neurons. Stimulus position and intensity were set to achieve 30−50% of maximus response amplitude in each cell. NMDAR EPSCs were pharmacologically isolated by adding 50 μM picrotoxin (Sigma-Aldrich, dissolved in DMSO) to block GABAergic transmission and DNQX (10 μM; Sigma-Aldrich, dissolved in DMSO) to block AMPA receptors. To evaluate the contribution of GluN2B containing-NMDARs, synaptic currents were evoked during 5 min to stablished a baseline response and after 10 min incubation in the bath solution with Ifenprodil 5 μM (Sigma Aldrich, France). For analysis, the time courses were obtained by normalizing each recording to the average value of all points constituting the first 5 min stable baseline. For the analysis the average of the traces were calculate in presence and absence of Ifenoprodil and the area under the curve (AUC) was calculated using Clampfit program. For the normalization per cell, the average of the amplitude of the currents were analyzed in presence and absence of Ifenoprodil. The data represent the percentage of change in presence of Ifenoprodil with respect the baseline in absence of antagonist.

For analyses of nAChR currents and nicotinic modulation of sEPSCs, nicotine was applied by a 'puff' with air-pressure pulses controlled by a Picospritzer III (General Valve). A drug-filled pipette was

moved within 20−40 μm from the recorded neuron and a pClamp protocol triggered a puff application of nicotine (10 mM) onto the recorded neuron with a 20−80 ms, 20-psi pressure ejection. Peak amplitude (pA) was compared between groups. Clampfit establishes whether the best fit of the traces should have 1 or 2 terms. This always corroborated visual observation. Time constants (τ values) of the decays were calculated by exponential fitting. In the cases where two phases were observed, we calculated the weighted τ (consisting of an exponential fit with two terms, one fast and one slow). This weighted time constant was calculated using the relative contribution from each of these components, applying the formula: τw = [(AF * τf) + (AS * τs)]/(AF + AS), where AF and AS are the relative amplitudes of the two exponential components, and τf and τs are the corresponding time constants. τ and τw were not different within the treatment groups, so they were pooled for between-groups comparison. Finally, charge transfer of nicotinic currents was calculated as the integral of the current (AUC) in the time window between onset and decay. To compare the frequency and amplitudes of sEPSCs in response to acute NIC exposure between treated animals and controls, epochs of at least 30 s were recorded before and after puff application of nicotine on VTA DA neurons. Analyses were performed using Clampfit 10.2 (Axon Instruments, USA) and Prism (Graphpad, USA).

### DREADD experiments

DAT-Cre mice, in which Cre recombinase expression is restricted to DA neurons without disrupting endogenous dopamine transporter (DAT) expression[63], were weaned at PND 21 ± 1 and treated with nicotine (100 μg/mL in 2% saccharine) for one week. When they reached adulthood, they were injected with Cre-dependent (DIO) DREADD or control (tdTomato) viruses. Adult DAT-Cre mice were anesthetized with a mixture of oxygen (1 L/min) and 3–4% isoflurane (Vetflurane, Virbac) for the induction of anesthesia, and then placed on a warming pad in a stereotaxic frame (David Kopf) and maintained under anesthesia throughout the surgery at 2% isoflurane. A local anesthetic (Lurocaine) was applied at the location of the scalp incision and 0.1 μL of buprenorphine (Buprecare, 1 mg/kg) was injected subcutaneously before the procedure. Mice then received bilateral injections of a retrogradely-transported inhibitory DREADD AAV (ssAAV-retro/2-hSyn1-dlox-hM4D(Gi)_mCherry(rev)-dlox-WPRE-hGHp(A), Viral Vector Facility Zurich) or control virus (ssAAV-retro/2-hEF1a-dlox-dTomato-EGFP(rev)-dlox-WPRE-hGHp(A), Viral Vector Facility Zurich) into the medial NAc (bregma +1.7 mm, lateral ±0.45 mm, ventral −4.1 mm).

Mice were tested in the EOM 4−5 weeks after viral injections to allow full expression of the DREADD or control viruses at the level of the VTA. Clozapine-N-oxide (5 mg/kg, CNO) was injected i.p. one hour before the mice received nicotine or saline and entered into the EOM task, as described above. While CNO can have off-target effects[78], the 5 mg/kg dose used in our experiments is below the threshold identified for off-target effects. In support of this idea, we did not see any differences in saline-injected mice under CNO, regardless of their virus (DREADD or control) status.

Following the EOM, mice were perfused with 1X PBS followed by 20 mL of 4% paraformaldehyde (Sigma). Brains were postfixed in PFA overnight, and transferred to PBS before sectioning. Serial 50 μm thick coronal sections were made on a Leica vibratome (VTS1000) within one week of perfusion, and stored free-floating in PBS. Free-floating brain sections were then incubated for 1 h at 4 °C in a blocking solution of phosphate-buffered saline (PBS) containing 3% bovine serum albumin (BSA, Sigma; A4503) (vol/vol) and 0.2% Triton X-100 (vol/vol), and then incubated overnight at 4 °C with a sheep anti-tyrosine hydroxylase antibody (anti-TH, Milipore, Ab1542) diluted 1:500 in PBS containing 1.5% BSA and 0.2% Triton X-100. The following day, sections were rinsed with PBS, and then incubated for 3 h at room temperature with Cy5-conjugated anti-sheep secondary antibody (Jackson ImmunoResearch, 713-175-147) diluted 1:500 in a solution of 1.5% BSA in PBS.

After three rinses in PBS, slices were wet-mounted using Prolong Gold Antifade Reagent containing DAPI (Invitrogen, P36931). Viral expression was evaluated by observing the unamplified expression of mCherry or tdTomato via a Rhodamine filter on an epifluorescent microscope (Zeiss Axio Imager), dopamine neurons were identified by TH staining via a Cy5 filter, and non-dopamine cell bodies were evaluated by DAPI staining. All neurons expressing the DREADD or control viruses co-expressed TH, and staining was limited to the medial VTA in line with previous reports on the location of neurons projecting to the medial NAc[79].

### Statistical analysis

All statistical analysis and graphs were made using R, a language and environment for statistical computing (Team, 2005, http://www.r-project.org), with the exception of in vitro electrophysiology data, which was analyzed in Prism (GraphPad). Data from behavioral experiments was exported from acquisition software (TSE system) or from ezTrack video tracking as a.csv file, and read directly into R. Quantifications from ClearMap or from bouton analysis in FIJI were likewise exported as.csv files and read directly into R. CSV files of cell counts by region were imported into R from ClearMap. Correlations and clustering were then performed in R using the {stats} package, and networks were created {igraph} package. For the measurement of neuronal activity in single-unit recordings, timestamps of action potentials were extracted in Spike 2, analyzed in R, and expressed as the average firing frequency (in Hz) and the percentage of spikes-within-burst (%SWB = number of spikes within burst divided by total number of spikes in a given window). Neuronal basal activity was defined on recordings of a minimum of three minutes. Firing frequency was quantified on overlapping 60-second windows shifted by 15-s time steps. For each neuron, the firing frequency was rescaled as a percentage of its baseline value averaged during 3 min before nicotine injection. The responses to nicotine are thus presented as a percentage of variation from baseline (mean ± S.E.M.). The effect of nicotine was assessed by comparison of the maximum of firing frequency variation induced by nicotine and saline injection. For activated (respectively inhibited) neurons, the maximal (respectively minimal) value of the firing frequency was measured within the response period (3 min) that followed nicotine or saline injection. The results are presented as mean ± S.E.M. of the difference of maximum variation after nicotine or saline.

### Reporting summary

Further information on research design is available in the Nature Portfolio Reporting Summary linked to this article.

## Data availability

Source data are provided with this paper. In addition, the data generated in this study have been deposited in the Zenodo database under https://doi.org/10.5281/zenodo.13684023 (https://zenodo.org/uploads/13684023). Source data are provided with this paper.

## Code availability

Graphs and statistical analyses were generated using standard code written in R (version 4.4.1). All codes used to run the analysis are available from the authors upon request.

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

## Acknowledgements

This work was supported by the Centre National de la Recherche Scientifique CNRS UMR 8246, INSERM U1130, the Foundation for Medical Research (FRM, Equipe FRM DEQ2013326488 to P.F.; Equipe FRM EQU202403018036 to J.B.), the French National Cancer Institute (Grant TABAC-16-022, TABAC-19-020 and SPA-21-002 to P.F.), and French state funds managed by the ANR (ANR-20 NICADO to P.F. and J.B., ANR-19-CE16-0028 Bavar to P.F. and N.R.). The project was also supported by the French government through the France 2030 investment plan managed by the National Research Agency (ANR), as part of the Initiative of Excellence of Université Côte d'Azur under reference number ANR-15-IDEX-01 and The Collegium of Advanced Studies to J.B. LMR was supported by a NIDA–Inserm Postdoctoral Drug Abuse Research Fellowship. C.N. was supported by a fourth-year PhD fellowship from the Biopsy Labex. We are grateful for support from the animal facilities (IBPS) and Otilia de Oliveira, Noemie Karakaplan-Dherbe and Emilie Tubeuf at ESPCI animal facilities.

## Author contributions

L.M.R. and P.F. designed the study. L.M.R., A.G., and C.F. performed the behavioral experiments. L.M.R., T.T., and DRajot performed iDISCO and ClearMap experiments with support from N.R. R.C.C., DRigoni and S.P.F. performed ex vivo electrophysiological recordings with support from J.B. L.M.R., S.L.F., C.N., T.L.B., and F.M. performed in vivo electrophysiological recordings. L.M.R. performed the surgeries and virus injections. L.M.R. and A.G. performed quantitative neuroanatomy experiments with support from N.H. L.M.R., A.G., R.C.C., DRigoni, T.T., J.B., and P.F. analyzed the data. L.M.R. wrote the paper with inputs from A.M., F.M., J.B., and P.F. All authors read and edited the manuscript. N.R., J.B., and P.F. secured the funding.

## Competing interests

The authors declare no competing interests.
