## [Peer Review File · Nature Communications]

REVIEWER COMMENTS

Reviewer #1 (Remarks to the Author):

This is a very interesting story shedding more light on adolescent nicotine exposure and its ramifications in adulthood. The authors present a very broad methodology, analyzing this complex issue on multiple levels. Though their results are novel and very compelling, I have several issues that I would like to raise.

1) The authors show that mice exposed to NIC in adolescence drink more nicotine than saccharin solution compared to those exposed to NIC in adulthood. To really make a statement that juvenile exposure to NIC increases the vulnerability to nicotine use it would be good to compare the nicotine consumption of adolescent vs. adult exposure between each other (NIC adolescent vs. NIC adulthood). It would be interesting to show that mice experiencing NIC in adolescence use more NIC than those exposed in adulthood. Based on the Sup Fig 1D that might be the case.

2) The authors suggest a juvenile-like “freezing” of glutamatergic receptor properties. And although the data points in that direction it is too early (in my opinion) to draw such a conclusion. There might be many reasons for reduced AMPA/NMDA ratios or different NMDARs currents kinetics. It would be great to support this statement by showing the difference in GluN2A/GluN2B subunit composition via recordings in the presence of ifenprodil. Without such experimental support, I would suggest toning down these rather strong conclusions.

3) In Fig3B, NIC puff elicits currents of similar peak amplitudes between SAC and NIC adolescent exposure groups. The representative traces however point to very different currents kinetics between these two groups. SAC current has a much slower decay time, hence much more current passes through the receptor, which can indicate a very different nicotine signaling. If those representative traces indeed reflect the currents in the whole group, perhaps showing this data as the charge transferred would better illustrate the results than the peak amplitude.

4) Fig 4C shows that chemogenetic inhibition of the VTA-NAc DA neurons induces such strong anxiety that the mouse didn't move beyond the closed arms of the O-maze. The authors later refer to it as restoring the adult-like behavior, while it much surpasses the adult-like behavior shown in Fig4A. What panel 4C does indicate is that when stimulated, the projection triggers severe anxiety-like behavior, hence NIC adolescent exposure does not irreversibly dull its function.

5) The discussion is very interesting yet often extrapolates the results of the manuscript to nicotine addiction development and vulnerability. Indeed the authors show important differences between adolescent and adult nicotine exposures on behavioral, cellular, and network levels. The paper however does not explore the ramifications of adolescent NIC exposure on the propensity to develop addiction. It would be very interesting to see if adolescent-exposed mice indeed develop compulsive nicotine use, present despite footshocks or other deterrents. But since the data do not go beyond a nicotine re-exposure (and not addiction-inducing behavioral models) I would advise the authors to refrain from making such far-reaching assumptions and conclusions (eg. the title of Fig 1).

6) iDISCO light-sheet results generate a vast amount of data that is difficult to represent in a short manuscript. Showing raw data and listing comparisons of c-Fos density for each detected structure between adolescent and adult-exposed mice would provide the scientific community with interesting information, giving targets, and answering questions not necessarily described and discussed in this manuscript. Perhaps making such data available on GitHub would be beneficial for the substance abuse research community in general.

Reviewer #2 (Remarks to the Author):

In this manuscript, the authors used a combination of behavioural analyses, chemogenetics and optogenetics along with electrophysiological analyses to examine the effects of transient exposure to oral nicotine administration during early adolescence on longer term sensitivity to nicotine reward and anxiety-related behavioural outcomes. They report that transient nicotine exposure in early adolescence was sufficient to produce a marked vulnerability to nicotine in adulthood, associated with disrupted functional connectivity in dopaminergic circuits. The mice showed persistent adolescent-like behavioral and physiological responses to nicotine, suggesting that nicotine exposure in adolescence somehow “froze” these circuits in an immature, imbalanced state. Next the authors suggest that chemogenetically resetting the balance between the underlying dopamine circuits is sufficient to reveal a mature behavioral response to acute nicotine in adolescent-exposed mice.

Overall the findings reported are very interesting and novel. I was impressed by the use of several cutting edge technologies to test the hypotheses in the paper.

Nevertheless, there were several limitations in the manuscript that limited its impact.

1. There are some concerns over the small n size for several of the neuronal electrophysiological recordings. In some cases, just two neurons were analysed making more global generalizations about the effects more limited in scope.

2. Concerns over the nicotine administration procedure: The use of sucrose is understandable given the difficulty in getting rodents to consume nicotine containing liquids. However, what might the sensitization effects of sucrose alone be in the reported effects? Certainly DA neurons are highly sensitive to sucrose reward consumption and there are some concerns that there is a potential effect conflation between the impacts of sucrose vs. nicotine on this highly sensitive circuitry.

3) I think its important to recognize the limitations of liquid nicotine consumption vs. human inhalation and discuss how the liquid consumption protocol in this paper may or may not correspond to human nicotine consumption patterns taking into account dosing differences and metabolic differences between rodents and humans. For example, what rationale was used for the selection of this specific nicotine dose regiment?

4) The use of a single behavioural output for anxiety limits the impact of the paper. For example, we know from the clinical and pre-clinical literature that depression and cognitive deficits are also common correlates of adolescent nicotine exposure. Thus, including even some simple behavioural correlates of anhedonia or cognitive deficits would have more convincingly demonstrated the overall impacts of the adolescent nicotine exposure. Similarly, it would have helped to measure nicotine reward or aversion sensitivity by running a CPP/CPA test. These tests are relatively quick and simple to perform in mice.

5) Related to this point, the main issue here is that the only outcome measures to make the claim that the DA system has somehow been suspended in an adolescent phenotype are the single anxiety outcome measures and the response analyses to nicotine exposure. It would have greatly strengthened the manuscript to have behavioural measures linked to "adolescent" phenotypes. For example, did they also show increased impulsivity? Less cognitive flexibility, etc? The DA circuit in question is really the central DA hub controlling many adolescent behavioural phenotypes so if nicotine exposure was indeed "freezing" the system in this immature state, one would expect other adolescent-related behavioural phenotypes to be present as well, to make a more convincing case for the central thesis of the paper.

6) Finally, the use of the words "freeze" and "thaw" throughout the entire manuscript seems inaccurate and not appropriate when referring to a dynamic, neurophysiological series of events. I

understand the colloquial attempt at the use of this language but it really makes no sense when referring to a living, biological series of events in a dynamic neural landscape. At best, there was evidence for the prevention of the "maturation" of this particular dopamine circuitry but that is not functionally equivalent to 'freezing' a neurophysiological system into some sort of static phenotype. I strongly encourage the authors to use alternative terminology in this context. For example, along the lines of "adolescent nicotine exposure kept the dopamine system in a persistent adolescent-like functional state" or something along those lines.

Reviewer #3 (Remarks to the Author):

In this manuscript, the authors examined the impact of nicotine exposure during adolescence or adulthood on later nicotine intake, VTA electrophysiological responses, c-fos brain expression, and anxiety (elevated plus maze). While the studies are of interest, they are incremental from other published reports in the field, and thus not particularly noteworthy. While the results of the whole-brain clearing are intriguing, they are confounded since subjects were examined following vehicle/nicotine AND EOM test; thus, it is unclear if the activity patterns based on adolescent drug history are due to anxiety induced with the EOM, acute nicotine injection, or an interaction of both factors. Additional concerns are as follows:

1. Some of the conclusions are not supported by the data as presented. For instance, it is indicated "replicated the pronounced, time-dependent reduction of time spent in the open arms of the EOM seen after nicotine administration in naive adult males, exposure to NIC in adolescence abolished this antigenic effect". First, there are no statistical analyses examining a time-dependent change; rather, the only statistics compare the treatment groups at each time point. Regarding exposure to NIC in adolescence, since the data are not directly compared across all 4 groups, this conclusion is confounded. Indeed, given the size of the error bars comparing Figure 1G, it doesn't appear that there is a significant difference between SAC in adolescence with acute nicotine, as compared to NIC in adolescence with either acute saline or nicotine. Similarly, all 4 groups should be compared together for each data set with Figure 1H, 4A and 4C.
2. It is not clear if the statistics corrected for multiple comparisons when the same set of data were analyzed more than once.

3. The contention that the circuits are 'frozen' by nicotine is not valid based on the data presented.

4. It is unclear if the heat map EOM images are inclusive of all subjects, or representative for only one subject in the group.

5. Figure 1B, this figure is difficult to understand and doesn't provide rigorous information about the study. It is unclear if this represents one subject, or mean of all the subjects. If mean, there is no SEM indicated to understand variability among subjects.

6. Similar to the above, there appears to be significant variability in the drinking and EOM data sets. Thus, it would be of benefit to represent each time point as a bar graph with individual subject data points indicated to better understand within group dynamics.

7. Images in the figures that outline the methods would be more appropriate in the supplementary material to allow the reader to better focus on the data.

8. The authors provide a nice rationalization for both reward or aversion being encoded in the VTA DA neurons, indicating that they are not a heterogeneous population. It is thus confusing that this factor was not apparently taken into consideration with the electrophysiology assessments.

In the following section, we address each of the comments raised by the reviewers. We highlight changes to the text in red throughout the manuscript file.

REVIEWER COMMENTS

Reviewer #1 (Remarks to the Author):

This is a very interesting story shedding more light on adolescent nicotine exposure and its ramifications in adulthood. The authors present a very broad methodology, analyzing this complex issue on multiple levels. Though their results are novel and very compelling, I have several issues that I would like to raise.

We thank the reviewer for their enthusiasm and their constructive comments. Please find our detailed responses below.

1) The authors show that mice exposed to NIC in adolescence drink more nicotine than saccharin solution compared to those exposed to NIC in adulthood. To really make a statement that juvenile exposure to NIC increases the vulnerability to nicotine use it would be good to compare the nicotine consumption of adolescent vs. adult exposure between each other (NIC adolescent vs. NIC adulthood). It would be interesting to show that mice experiencing NIC in adolescence use more NIC than those exposed in adulthood. Based on the Sup Fig 1D that might be the case.

First, it is important to clarify that in our oral consumption experiments mice were always offered the choice between nicotine (in plain water) or water – there was never saccharin in the two-bottle choice solutions. This is in contrast to the initial, forced, pre-treatment, where we added saccharin to counteract the bitter flavor of the nicotine. Adding saccharin thus reduces the risk that mice will not drink enough liquid due to the flavor of the nicotine, which can be dangerous for the adolescent mice in particular.

As for the comparison between mice treated with nicotine in adolescence or adulthood, the validity of this comparison is necessarily limited by our experimental design. First, experiments were performed separately for each pre-treatment age. Because experiments within each pre-treatment age always contained both nicotine or saccharin treated mice, we can confirm that these results do not vary significantly between different cohorts tested at different times. We cannot confirm this for the adult x adolescent comparison. Secondly, because the period between pre-treatment and testing was kept the same for each pre-treatment age, the mice are not the same age at the moment of adult testing (with the adult-treated mice ~ 5 weeks older than adolescent-treated mice). We thus consider that the most appropriate and conservative statistical approach is to compare the nicotine-treated groups only to their controls treated with saccharin at the same age. Taking these caveats into account, we cannot conclude that mice exposed to NIC in adolescence drink more nicotine solution than water when compared to those exposed to NIC in adulthood. Our conclusion is that mice exposed to NIC during adolescence show a modification of their drinking behavior in adulthood when compared to their age-matched, control-treated congeners that is not apparent if the mice are exposed in adulthood. This is clearly specified in the text to avoid any ambiguity.

Lines 79-85: “Adult mice exposed to NIC in adolescence consumed more nicotine solution as a percentage of their total liquid intake than their congeners exposed only to SAC in adolescence at all doses tested, (Fig 1B *middle*), with differences in the daily dose of nicotine most obvious at the 50 µg/ml and 100 µg/ml doses. This did not result from side bias in consumption, as all mice were able to track the position of the nicotine or sucrose solution (Sup Fig 1C). In stark contrast, mice exposed to NIC in adulthood did not show different consumption behavior from their SAC-treated counterparts (Fig 1C), suggesting that this 1-week pretreatment regimen is too mild to produce enduring changes to the adult brain.”

2) The authors suggest a juvenile-like “freezing” of glutamatergic receptor properties. And although the data points in that direction it is too early (in my opinion) to draw such a conclusion. There might be many reasons for reduced AMPA/NMDA ratios or different NMDARs currents kinetics. It would be great to support this statement by showing the difference in GluN2A/GluN2B subunit composition via recordings in the presence of ifenprodil. Without such experimental support, I would suggest toning down these rather strong conclusions.

We thank the reviewer for this suggestion. We have now performed a new set of electrophysiology experiments in order to assess NMDA currents in the presence of ifenprodil. We did not find strong evidence for the presence of NR2B receptors specifically in the NIC-treated group in these initial experiments. Further experiments would thus

be needed to confirm or to rule out the contribution of these receptors to the plasticity changes that we observed in the VTA. For the interest of this manuscript, we have decided to save these experiments for future projects, as these results, while interesting, are not directly related to the main message of the paper – which is, that nicotine exposure in adolescence prolongs an adolescent-like neurophysiological and behavioral response to re-exposure in adult animals, primarily through alterations in VTA-NAc signaling in response to nicotine. In accordance, we have made the following changes to the figures and text as to not misrepresent our data.

Please see the results of this experiment in Supplementary Figure 3F

See Figure Legend (Supplementary Information, Lines 398-400): “(F) Average change in NMDAR currents following application of Ifenprodil (5 mM) to VTA DA neurons (SAC N = 4, n = 7; NIC N = 4, n = 7).”

See Results (Lines 170 - 175) : “These mice showed a reduction in AMPA/NMDA ratio in comparison to SAC-treated controls (Fig 3C *left*) and an increase in the weighted NMDA decay time constant (τ_w , Fig 3C *right*), **albeit without a clear contribution of changes to NR2B subunit expression (Sup Fig 3F)**. Interestingly, these alterations to glutamatergic plasticity onto VTA dopamine neurons of adult mice exposed to NIC as adolescents **resemble some conditions** observed in young brains^{16,41–44}, suggesting that NIC exposure in adolescence **may impede the maturation of glutamatergic plasticity mechanisms.**”

See Supplemental Methods (Supplementary Information, lines 223-237): “To assess NMDA Currents, voltage-clamp were performed using an internal solution containing (in mM) 130 CsCl, 4 NaCl, 2 MgCl₂, 1.1 EGTA, 5 HEPES, 2 Na₂ATP, 5 sodium creatine phosphate, 0.6 Na₃GTP, and 0.1 spermine. Cells were held at +40mV and NMDA synaptic currents were evoked 15 times (3 times per min) by stimuli of 0.1ms every 20 s through a glass pipette placed 200 μ m from the patched neurons. Stimulus position and intensity were set to achieve 30-50% of maximum response amplitude in each cell. NMDAR EPSCs were pharmacologically isolated by adding 50 μ M picrotoxin (Sigma-Aldrich, dissolved in DMSO) to block GABAergic transmission and DNQX (10 μ M; Sigma-Aldrich, dissolved in DMSO) to block AMPA receptors. To evaluate the contribution of GluN2B containing-NMDARs, synaptic currents were evoked during 5 min to establish a baseline response and after 10 min incubation in the bath solution with Ifenprodil 5 μ M (Sigma Aldrich, France). For analysis, the time courses were obtained by normalizing each recording to the average value of all points constituting the first 5 min stable baseline. For the analysis the average of the traces were calculate in presence and absence of Ifenprodil and the area under the curve (AUC) was calculated using Clampfit program. For the normalization per cell, the average of the amplitude of the currents were analyzed in presence and absence of Ifenprodil. The data represent the percentage of change in presence of Ifenprodil with respect the baseline in absence of antagonist.”

3) In Fig3B, NIC puff elicits currents of similar peak amplitudes between SAC and NIC adolescent exposure groups. The representative traces however point to very different currents kinetics between these two groups. SAC current has a much slower decay time, hence much more current passes through the receptor, which can indicate a very different nicotine signaling. If those representative traces indeed reflect the currents in the whole group, perhaps showing this data as the charge transferred would better illustrate the results than the peak amplitude.

Based on the reviewer comments, we went back to see whether the traces we chose were indeed representative. They are representative. We then performed additional analyses by assessing the quality of fit of all of the traces, and found that neurons showed 2 types of responses to nicotine: monophasic (one component) and bi-phasic (with a fast and slow component). We then fitted different equations based on the type of responses to assess the kinetics of desensitisation and extract the time constant τ . In the cases where two phases were observed we calculated the weighted τ (consisting of an exponential fit with two terms, one fast and one slow). This weighted time constant was calculated using the relative contribution from each of these components, applying the formula: $\tau_w = [(af * \tau_f) + (as * \tau_s)] / (af + as)$, where af and as are the relative amplitudes of the two exponential components, and τ_f and τ_s are the corresponding time constants. τ and τ_w within the groups. Finally, charge transfer of nicotinic currents was calculated as the integral of the current (AUC) in the time window between onset and decay.

We discovered that adult mice treated with NIC in adolescence were more likely to show a biphasic response than those treated with SAC in adolescence (7/14 neurons, vs. 3/13 in the SAC group). Despite showing the same current amplitude, adult mice exposed to nicotine in adolescence showed a faster return to baseline after a nicotine puff, indicating a difference in current kinetics between the two groups. This kinetic difference indeed leads to a difference in the charge transferred.

We have now added this data to Supplementary Figure 3E

See Figure Legend (Supplementary Information, Lines 393-398)

“(E) *Left* : Nicotine current traces were analyzed and neurons were found to show two types of responses to nicotine: monophasic (one component) and bi-phasic (with a fast and slow component). More neurons from adult mice pretreated with NIC in adolescence showed a biphasic response than those treated with SAC. *Center*: Nicotine currents in neurons from mice treated with NIC in adolescence showed a lower τ indicative of a faster current decay. *Right*: Less charge is transferred in response to a nicotine puff in neurons from mice treated with NIC in adolescence.”

See Results Lines 164-167: “The peak amplitude of nicotine currents did not differ between mice exposed to NIC or SAC in adolescence (Fig 3B *left*) or adulthood (Fig 3B *right*), however NIC treated mice showed faster response kinetics and an associated reduction in charge transferred (Sup Fig 3E), suggesting that signaling through nAChR receptors may be altered in the VTA of these mice.”

See Supplemental Methods (Supplementary Information, lines 243-251): “Clampfit establishes whether the best fit of the traces should have 1 or 2 terms. This always corroborated visual observation. Time constants (τ values) of the decays were calculated by exponential fitting. In the cases where two phases were observed, we calculated the weighted τ (consisting of an exponential fit with two terms, one fast and one slow). This weighted time constant was calculated using the relative contribution from each of these components, applying the formula: $\tau_w = [(AF * \tau_f) + (AS * \tau_s)] / (AF + AS)$, where AF and AS are the relative amplitudes of the two exponential components, and τ_f and τ_s are the corresponding time constants. τ and τ_w were not different within the treatment groups, so they were pooled for between-groups comparison. Finally, charge transfer of nicotinic currents was calculated as the integral of the current (AUC) in the time window between onset and decay.”

4) Fig 4C shows that chemogenetic inhibition of the VTA-NAc DA neurons induces such strong anxiety that the mouse didn't move beyond the closed arms of the O-maze. The authors later refer to it as restoring the adult-like behavior, while it much surpasses the adult-like behavior shown in Fig4A. What panel 4C does indicate is that when stimulated, the projection triggers severe anxiety-like behavior, hence NIC adolescent exposure does not irreversibly dull its function.

We thank the reviewer for bringing this point to our attention. Of note, we are not stimulating the VTA-NAc pathway in this experiment, rather dampening its activity using an inhibitory DREADD approach. This approach has been shown to decrease the excitability of VTA DA neurons, which should render them less sensitive to the effects of nicotine (see Bariselli et al 2018).

Importantly, our results in panel 4D indicate that the DREADD approach we used in this experiment produced an effect only when it was paired with the nicotine injection, as the hM4D(Gi)-SAL (i.e. DREADD+CNO+saline) treated mice did not show an anxiogenic or anxiolytic effect different from other saline-treated groups. We concluded (lines 276-279) that “The DREADD + nicotine mice were the only group tested that showed a significant anxiogenic response in the EOM, indicating that decreasing the excitability of VTA-NAc DA neurons restored the mature behavioral response to nicotine, effectively unmasking the anxiogenic effect of the drug.”, We wish to emphasize that we restore the anxiogenic effect, and do not specifically assess the *degree* of this effect.

We now take the opportunity to clarify these points in the main text of the paper:

Lines 238-248 now read : “In contrast, dampening VTA-NAc DA activity with the DREADD virus caused the mice to spend dramatically less time in the open arms of the EOM after i.p. nicotine injection (Figure 4D *right*), an effect closely resembling the response to nicotine in naive adults (Fig 4B) and in adult mice treated with SAC in adolescence (Fig 1E). However, both mCherry and DREADD injected mice that received CNO one hour before being tested with saline in the EOM showed similar levels of exploration of the open arms, in line with our other experiments, indicating that (1) exposure to the CNO *itself* does not produce an anxiogenic or anxiolytic effect, and (2) decreasing the excitability of the VTA-NAc pathway alone (i.e. the hM4D(Gi)-SAL condition) does not produce an anxiogenic or anxiolytic effect. The DREADD + nicotine mice were the only group tested that showed a significant anxiogenic response in the EOM, indicating that decreasing the excitability of VTA-NAc DA neurons restored the mature behavioral response to nicotine, effectively unmasking the anxiogenic effect of the drug.”

Lines 251-252 now read: “Furthermore, our findings from the DREADD experiments indicate that this imbalance can be rescued by targeted pathway manipulations.”

5) The discussion is very interesting yet often extrapolates the results of the manuscript to nicotine addiction development and vulnerability. Indeed the authors show important differences between adolescent and adult nicotine exposures on behavioral, cellular, and network levels. The paper however does not explore the ramifications of adolescent NIC exposure on the propensity to develop addiction. It would be very interesting to see if adolescent-exposed mice indeed develop compulsive nicotine use, present despite footshocks or other deterrents. But since the data do not go beyond a nicotine re-exposure (and not addiction-inducing behavioral models) I would advise the authors to refrain from making such far-reaching assumptions and conclusions (eg. the title of Fig 1).

We now take the opportunity to clarify in the text that we do not address directly addiction-inducing behavioral models, but instead focus on the balance between the reinforcing and anxiety-inducing effects of nicotine exposure which has been posited to play a role in the first steps toward addiction.

To this end, we have changed the title of Figure 1 to “*Brief exposure to nicotine in adolescence induces a life-long imbalance between the rewarding and anxiogenic effects of later exposure*” which is more descriptive of the findings without evoking the concept of addiction.

In addition, lines 113-115 now read: “Together, these results define an adolescent period where exposure to nicotine produces an enduring profile of altered response to later nicotine, featuring a reduction of its anxiogenic properties, **an increased sensitivity to its reinforcing properties**, and an increase in voluntary consumption consistent **with a vulnerable phenotype**.”

Finally, we want to add that we also performed a series of new experiments, where we tested the sensitivity of each group of mice in the manuscript (naïve adults, naïve adolescents, adult mice treated with nicotine or saccharin as adolescents, or adult mice treated with nicotine or saccharin as adults) to nicotine conditioned place preference with a low dose of the drug. These experiments confirm and reinforce our results. While they do not introduce the concept of compulsive drug-taking behavior, these results bolster the idea that both the reinforcing and anxiety-inducing effects of nicotine exposure are modified by adolescent exposure to nicotine, leading to an imbalance in response which may represent a vulnerability in the first steps toward developing an addiction-like profile.

6) iDISCO light-sheet results generate a vast amount of data that is difficult to represent in a short manuscript. Showing raw data and listing comparisons of c-Fos density for each detected structure between adolescent and adult-exposed mice would provide the scientific community with interesting information, giving targets, and answering questions not necessarily described and discussed in this manuscript. Perhaps making such data available on GitHub would be beneficial for the substance abuse research community in general.

We thank the reviewer for their enthusiasm on this data set. Indeed, we can confirm that all the data for the paper, as well the codes used to analyse and make figures, will be available on our lab github at the acceptance of the paper for publication. We look forward to hearing from our colleagues whether the extended iDISCO data set may provoke interesting questions for them to follow.

Reviewer #2 (Remarks to the Author):

In this manuscript, the authors used a combination of behavioural analyses, chemogenetics and optogenetics along with electrophysiological analyses to examine the effects of transient exposure to oral nicotine administration during early adolescence on longer term sensitivity to nicotine reward and anxiety-related behavioural outcomes. They report that transient nicotine exposure in early adolescence was sufficient to produce a marked vulnerability to nicotine in adulthood, associated with disrupted functional connectivity in dopaminergic circuits. The mice showed persistent adolescent-like behavioral and physiological responses to nicotine, suggesting that nicotine exposure in adolescence somehow “froze” these circuits in an immature, imbalanced state. Next the authors suggest that chemogenetically resetting the balance between the underlying dopamine circuits is sufficient to reveal a mature behavioral response to acute nicotine in adolescent-exposed mice. Overall the findings reported are very interesting and novel. I was impressed by the use of several cutting edge technologies to test the hypotheses in the paper.

We thank the reviewer for their positive comments and constructive critiques. Please see below for our point-by-point responses.

Nevertheless, there were several limitations in the manuscript that limited its impact.

1. There are some concerns over the small n size for several of the neuronal electrophysiological recordings. In some cases, just two neurons were analysed making more global generalizations about the effects more limited in scope.

To clarify, the N indicated for electrophysiological experiments refers to the number of mice tested in the experiment, while the n refers to the number of neurons evaluated. Thus, the experiment in question tested 13-14

neurons from 2 mice. We have verified that this information appears in the figure (e.g. N = 2, n = 14), but also is clearly indicated in the figure legend (e.g. NIC N = 2 mice, n = 14 neurons) for patch experiments, and we have clarified this point by adding the N and n to the figure legend for in vivo electrophysiology experiments, while previously they appeared only in the figure itself.

2. Concerns over the nicotine administration procedure: The use of sucrose is understandable given the difficulty in getting rodents to consume nicotine containing liquids. However, what might the sensitization effects of sucrose alone be in the reported effects? Certainly DA neurons are highly sensitive to sucrose reward consumption and there are some concerns that there is a potential effect conflation between the impacts of sucrose vs. nicotine on this highly sensitive circuitry.

We thank the reviewer for bringing up this important topic. While it is true that sucrose can affect dopamine neurons, in the present study we exclusively use a 2% saccharin solution to mask the taste of nicotine only in the pre-treatment solutions. Because saccharin is non-nutritive, it has been suggested to affect dopamine neurons to a lesser extent than sucrose (McCutcheon 2015, *Physiology & Behavior*). Importantly, because we could not rule out the possibility that exposure to saccharin in adolescence would have enduring effects on dopaminergic function and response to later nicotine, a saccharin-only treated control was always used, and groups were always tested at the same time (i.e. each day of each experiment treated nicotine- and saccharin- pre-treated groups in parallel.) Finally, we saw results in adult mice pre-treated with saccharin in adolescence that closely matched the results in naïve adult mice, further suggesting that this control solution did not significantly alter adolescent development.

3) I think its important to recognize the limitations of liquid nicotine consumption vs. human inhalation and discuss how the liquid consumption protocol in this paper may or may not correspond to human nicotine consumption patterns taking into account dosing differences and metabolic differences between rodents and humans. For example, what rationale was used for the selection of this specific nicotine dose regimen?

We thank the reviewer for the opportunity to clarify this point in the manuscript. We do not attempt with this study to make a direct correlate between human and animal consumption patterns. Humans that begin using nicotine in adolescence are notably more likely to continue use into adulthood, leading to significant confounding effects of age of onset and time spent using nicotine. Instead, in this study we designed the exposure regimen with the aim to confine nicotine consumption to a defined developmental period in order to assess the enduring consequences on neural circuitry. The nicotine dose selected was based on the existing literature as to nicotine drinking behavior in C57 mice which indicates that it is a medium-to-high concentration that is readily consumed in the majority of mice (e.g. Klein et al., *Pharmacol Biochem Beh* 2004; Bagdas et al., *Neuropharmacology* 2019; Mondoloni et al., *eLife* 2023), and that it produces serum cotinine levels in mice (Klein et al., *Pharmacol Biochem Beh* 2019) that are in line with serum cotinine levels seen in adolescent nicotine users (Branstetter and Muscat, *Nicotine Tob. Res* 2013).

We have now added following to the discussion (Lines 319-324) to address this point: “The oral nicotine consumption paradigm used in the current studies cannot be considered a direct model of adolescent nicotine use in humans, notably as the route of nicotine administration (and thus its pharmacokinetic properties⁵⁷), the dosing, and the metabolism of the drug differ between species. However, here we are able to show that exposure to nicotine during a discrete period in adolescence locks the function of DA neurons in a persistent adolescent-like state, leading to exaggerated response to the rewarding effects of nicotine and a blunting of its negative, anxiogenic effects.”

4) The use of a single behavioural output for anxiety limits the impact of the paper. For example, we know from the clinical and pre-clinical literature that depression and cognitive deficits are also common correlates of adolescent nicotine exposure. Thus, including even some simple behavioural correlates of anhedonia or cognitive deficits would have more convincingly demonstrated the overall impacts of the adolescent nicotine exposure. Similarly, it would have helped to measure nicotine reward or aversion sensitivity by running a CPP/CPA test. These tests are relatively quick and simple to perform in mice.

We now take this opportunity to reiterate that the outcomes assessed in the current manuscript are focused on evaluating how exposure to nicotine in adolescence alters neural circuitry to influence later response to nicotine. Assessing other behavioral outcomes, such as cognitive deficits, was not the goal of this manuscript and as such they are not discussed. These outcomes, however, have been evaluated in other publications. For example, Fountain et al., *Exp Brain Res* 2008, and Pickens et al., *Neurotoxicol Teratol* 2013 showed that male rats exposed to nicotine in adolescence showed deficits on a Serial Pattern Learning task associated with a slower learning of the task. Counette et al., *Neuropsychopharmacol* 2009 and *Nat Neurosci* 2011 showed that adult rats exposed to

nicotine in adolescence showed diminished attentional performance and increased impulsive action in the Five-Choice Serial Reaction Time Task.

In lines 259-261 of the discussion, we now clarify: “We propose that exposure to nicotine in adolescence prolongs a developmental imbalance in dopaminergic circuitry **and in behavioral response to nicotine re-exposure**, which, in turn, can promote vulnerability to nicotine use and addiction.”

We do however agree with the reviewer that our evaluation of nicotine response has been narrowly focused on the anxiogenic effects of the drug. We performed a series of new experiments based on this reviewer’s suggestion, where we tested the sensitivity of each group of mice in the manuscript (naïve adults, naïve adolescents, adult mice treated with nicotine or saccharin as adolescents, or adult mice treated with nicotine or saccharin as adults) to nicotine conditioned place preference with a low dose of the drug. In our previous work (Durand-de Cuttoli et al., *eLife* 2018, Jehl et al *bioRxiv* 2023) we have shown that naïve adult male mice form nicotine CPP to a 0.5mg/kg dose of nicotine, but not to a 0.2 mg/kg dose of nicotine. In the new experiments for the current manuscript, we tested all groups, over 100 mice, for nicotine CPP at a 0.2 mg/kg dose of nicotine. In line with our previous results, naïve adult male mice did not show nicotine CPP (Figure 4A), while in line with the literature (Kota et al., *J. Pharmacol. Exp. Ther.* 2007, *Biochemical Pharmacology* 2009; Vastola et al., *Physiology & Behavior* 2002), naïve adolescent male mice showed a robust CPP response (Fig 4A). Excitingly, we found that our adult mice treated with nicotine in adolescence again showed an adolescent-like response with a robust CPP response to 0.2 mg/kg of nicotine in comparison to their SAC-treated counterparts (Figure 1F). Mice treated with NIC or SAC as adults showed no CPP (Figure 1I). The addition of these new data strengthen our message by highlighting that a behavioral imbalance in nicotine response is present in adolescent animals and persists into adulthood after adolescent nicotine exposure, in line with our electrophysiological data.

Please see the new data in Figures 1F (CPP in adult mice treated as adolescents), 1I (CPP in adult mice treated as adults), 4A (CPP in adolescent and adult naïve mice).

Figure legend 1 now reads (lines 371-373) “**(F) Adult mice treated with SAC in adolescence did not show CPP to a 0.2 mg/kg dose of nicotine, which is known not to induce CPP in naïve adult animals. Adult mice treated with NIC in adolescence, however, showed CPP to this low dose of nicotine.**”, and (line 378) “**(I) Mice treated with NIC or SAC as adults do not show CPP to a low dose of nicotine.**”

Figure legend 4 now reads (lines 436-438) “. **(A) Experimental timeline for naïve mice (left). Following a 5-day CPP paradigm, adolescent mice showed a significant place preference for the chamber paired with 0.2mg/kg nicotine (center), while adult mice did not (right).**”

Lines 105-111 of the Results now reads : “**We have previously shown that adult male mice show CPP to a 0.5mg/kg dose of nicotine³¹, but not to a 0.2 mg/kg dose of nicotine³². Here, we tested whether adult mice treated with nicotine in adolescence were more sensitive to the reinforcing effects of a low dose of nicotine. Mice exposed to SAC in adolescence did not show CPP to a 0.2 mg/kg dose of nicotine, in accordance with our previous results in naïve adult mice. However, mice treated with NIC in adolescence showed a significant preference for the nicotine-paired chamber at this same dose (Fig 1F), suggesting that these mice are more sensitive to the reinforcing effects of nicotine. Mice treated with NIC or with SAC as adults showed no preference for the nicotine-paired chamber (Fig 1I).**”

Lines 212-217 of the Results now reads : “**Naïve adolescent or adult mice underwent a 5-day CPP paradigm where either one compartment was paired with saline and one with 0.2 mg/kg of nicotine, or both chambers were paired with saline. Multiple previous studies have shown that adolescent rodents show CPP to lower doses of nicotine than adult rodents do⁴⁵⁻⁴⁷. We indeed found that adolescent mice showed a robust place preference to nicotine (Fig 4A center), while adult mice showed no preference for the nicotine-paired chamber (Fig 4A right) , in line with our previous findings³².**”

5) Related to this point, the main issue here is that the only outcome measures to make the claim that the DA system has somehow been suspended in an adolescent phenotype are the single anxiety outcome measures and the response analyses to nicotine exposure. It would have greatly strengthened the manuscript to have behavioural measures linked to "adolescent" phenotypes. For example, did they also show increased impulsivity? Less cognitive flexibility, etc? The DA circuit in question is really the central DA hub controlling many adolescent behavioural phenotypes so if nicotine exposure was indeed "freezing" the system in this immature state, one would expect other adolescent-related behavioural phenotypes to be present as well, to make a more convincing case for the central thesis of the paper.

Please see response to the above, related, point.

6) Finally, the use of the words “freeze” and “thaw” throughout the entire manuscript seems inaccurate and not appropriate when referring to a dynamic, neurophysiological series of events. I understand the colloquial attempt at the use of this language but it really makes no sense when referring to a living, biological series of events in a dynamic neural landscape. At best, there was evidence for the prevention of the “maturation” of this particular dopamine circuitry but that is not functionally equivalent to ‘freezing’ a neurophysiological system into some sort of static phenotype. I strongly encourage the authors to use alternative terminology in this context. For example, along the lines of “adolescent nicotine exposure kept the dopamine system in a persistent adolescent-like functional state” or something along those lines.

We thank the reviewer for sharing this concern with us, we have removed all uses of the word “thaw” and all but one uses of the word “freezes” in the text, and where we have kept it we have included a detailed explanation of what we mean by this term. We have made the following changes to the text to address this point :

- Lines 24-26 now read: “These mice showed persistent adolescent-like behavioral and physiological responses to nicotine, suggesting that nicotine exposure in adolescence prolongs an immature, imbalanced state in the function of these circuits.”
- Lines 205-208 now read: “Our electrophysiological results raise the intriguing hypothesis that exposure to nicotine in adolescence “freezes” dopamine circuitry in an imbalanced, immature state, which promotes vulnerability to nicotine use. More concretely, this means that exposure to nicotine in adolescence prevented the maturation of the dopamine system, keeping its function in a persistent adolescent-like state.”
- Lines 223-227 now read: “As this strongly resembled the behavioral effect that we observed in mice treated with NIC in adolescence (Fig 1G), this result suggests that not only does nicotine in adolescence block both neural and behavioral responses to nicotine in an immature state, but that imbalance between nicotine-induced activation and inhibition of VTA dopamine neurons could be actively masking the anxiogenic effect of the drug.”
- Lines 274-276 now read: “Here, our evidence suggests that nicotine in adolescence arrests the development of VTA-NAc and VTA-AMg DA circuits, keeping them in a prolonged adolescent state.”
- Lines 283-284 now read: “Importantly, we also provide the first evidence that adult-like behavior can be restored by artificially re-calibrating neural responses to adult levels.”
- Lines 325-327 now read: “Because we were able to restore an adult-like behavioral response to nicotine with a chemogenic approach, our findings suggest that these effects of nicotine in adolescence may be reversible.”

Reviewer #3 (Remarks to the Author):

In this manuscript, the authors examined the impact of nicotine exposure during adolescence or adulthood on later nicotine intake, VTA electrophysiological responses, c-fos brain expression, and anxiety (elevated plus maze). While the studies are of interest, they are incremental from other published reports in the field, and thus not particularly noteworthy. While the results of the whole-brain clearing are intriguing, they are confounded since subjects were examined following vehicle/nicotine AND EOM test; thus, it is unclear if the activity patterns based on adolescent drug history are due to anxiety induced with the EOM, acute nicotine injection, or an interaction of both factors. Additional concerns are as follows:

1. Some of the conclusions are not supported by the data as presented. For instance, it is indicated “replicated the pronounced, time-dependent reduction of time spent in the open arms of the EOM seen after nicotine administration in naïve adult males, exposure to NIC in adolescence abolished this antigenic effect”. First, there are no statistical analyses examining a time-dependent change; rather, the only statistics compare the treatment groups at each time point. Regarding exposure to NIC in adolescence, since the data are not directly compared across all 4 groups, this conclusion is confounded. Indeed, given the size of the error bars comparing Figure 1G, it doesn't appear that there is a significant difference between SAC in adolescence with acute nicotine, as compared to NIC in adolescence with either acute saline or nicotine. Similarly, all 4 groups should be compared together for each data set with Figure 1H, 4A and 4C.

Indeed, the time-dependent effect of nicotine on anxiety-like behavior was not explicitly assessed in the manuscript. We have now edited this passage (Lines 93-100) to read: “We have previously found that naïve adult male mice show a pronounced, time-dependent reduction of time spent in the open arms of the EOM following nicotine administration²³. Here, we found that mice exposed to SAC in adolescence resembled our results in naïve mice: these mice showed an anxiogenic response to nicotine, as evidenced by a significant reduction in time spent in the

open arms in the 6-to-9-minute block when compared to saline-injected counterparts. There was, however, no difference in time spent in the open arms in the 6-to-9-minute block between saline and nicotine injected mice that were previously exposed to nicotine in adolescence, indicating that exposure to NIC in adolescence abolished this anxiogenic effect (Fig 1E).”

In accordance with our previous results (Nguyen et al., *Neuron* 2021), we found that mice that experienced nicotine for the first time in the EOM environment (i.e. mice that were previously exposed to saccharin in adolescence or in adulthood) indeed showed a significant time-dependent reduction of time spent in the open arms of the EOM (Kruskal-Wallis $\chi^2(2) = 10.723$, $p = 0.004694$, see associated figure). This effect was not evident in mice receiving saline injections (Kruskal-Wallis $\chi^2(2) = 0.97215$, $p = 0.615$).

To further clarify the subject of our statistical analysis, as the results were not normally distributed all four treatment groups were first compared with a Kruskal-Wallis Rank Sum Test within each time point for the manuscript, as indicated in Table S1. Between-group differences in the 6-to-9-minute bin were then broken down with repeated Wilcoxon tests. These tests were always corrected for multiple comparisons with a Holm correction. With regards to the clarity of the statistical analyses, we have now included the following text in the legends of Figures 1 and 4 to clearly indicate where group comparisons were made: “Graphs are separated by pre-treatment group for clarity, but statistical analyses compared all four treatment conditions.”

2. It is not clear if the statistics corrected for multiple comparisons when the same set of data were analyzed more than once.

It is indicated on the Tables S1-S4 when statistical analyses were corrected for multiple comparisons. However, to improve clarity, we have now added the following text to the legends of figures 1 and 4 “Holm’s sequential Bonferroni corrections were used to correct for multiple comparisons.”

3. The contention that the circuits are 'frozen' by nicotine is not valid based on the data presented.

We thank the reviewer for sharing this concern with us, we have removed all uses of the word “thaw” and all but one uses of the word “freezes” in the text, and where we have kept it we have included a detailed explanation of what we mean by this term. We have made the following changes to the text to address this point :

- Lines 24-26 now read: “These mice showed persistent adolescent-like behavioral and physiological responses to nicotine, suggesting that nicotine exposure in adolescence prolongs an immature, imbalanced state in the function of these circuits.”
- Lines 205-208 now read: “Our electrophysiological results raise the intriguing hypothesis that exposure to nicotine in adolescence “freezes” dopamine circuitry in an imbalanced, immature state, which promotes vulnerability to nicotine use. More concretely, this means that exposure to nicotine in adolescence prevented the maturation of the dopamine system, keeping its function in a persistent adolescent-like state.”
- Lines 223-227 now read: “As this strongly resembled the behavioral effect that we observed in mice treated with NIC in adolescence (Fig 1G), this result suggests that not only does nicotine in adolescence block both neural and behavioral responses to nicotine in an immature state, but that imbalance between nicotine-induced activation and inhibition of VTA dopamine neurons could be actively masking the anxiogenic effect of the drug.”
- Lines 274-276 now read: “Here, our evidence suggests that nicotine in adolescence arrests the development of VTA-NAc and VTA-AMg DA circuits, keeping them in a prolonged adolescent state.”
- Lines 283-284 now read: “Importantly, we also provide the first evidence that adult-like behavior can be restored by artificially re-calibrating neural responses to adult levels.”
- Lines 325-327 now read: “Because we were able to restore an adult-like behavioral response to nicotine with a chemogenic approach, our findings suggest that these effects of nicotine in adolescence may be reversible.”

4. It is unclear if the heat map EOM images are inclusive of all subjects, or representative for only one subject in the group.

We thank the reviewer for pointing out this oversight. We now specify in the legends for figures 1 and 4 that the heatmaps shown are from representative individual animals (Lines 379 and 451).

5. Figure 1B, this figure is difficult to understand and doesn't provide rigorous information about the study. It is unclear if this represents one subject, or mean of all the subjects. If mean, there is no SEM indicated to understand variability among subjects.

This representation gives an overview of drinking activity per solution per hour per day, indicating that (1) mice drink more in their active hours (dark cycle) than in their inactive times, as expected, and (2) pretreatment with nicotine or saccharin in adolescence does not alter this diurnal dynamic typical of mice. In line with the critique of this reviewer, we have now moved these actograms to Supplementary Figure 1B, and we now specify in the legend that shaded regions in the actogram represent mean volumes across the groups for each solution tested. This representation of the data is not well suited to exploring the variability between subjects, but see supplementary figure 1E, 1F for additional data on individual differences in drinking behavior.

6. Similar to the above, there appears to be significant variability in the drinking and EOM data sets. Thus, it would be of benefit to represent each time point as a bar graph with individual subject data points indicated to better understand within group dynamics.

Individual subject points are always present on all data represented in bar graphs or box plots, including the recently added CPP data. For measurements taken over time (e.g. in the two bottle choice and O maze experiments), the data are presented as grouped line graphs in the main figures for ease of visibility. Plots with the data shown for individual mice have now been added to Supplementary Figure 1 (see Supplementary figure 1E, 1F for the two bottle choice data, and Supplementary figure 1H, 1J for the O Maze data), as we have previously done (Nguyen et al., *Neuron* 2021).

7. Images in the figures that outline the methods would be more appropriate in the supplementary material to allow the reader to better focus on the data.

We thank the reviewer for sharing their opinion on the methodological parts of the figures. However, based on the feedback of other reviewers and previous readers of the manuscript we have decided to keep the method outlines in the figures, while reducing their size as much as possible without sacrificing readability. For readers less familiar with developmental studies these figures serve to quickly orient their attention to the age and treatment status of the animals used in each experiment.

8. The authors provide a nice rationalization for both reward or aversion being encoded in the VTA DA neurons, indicating that they are not a heterogeneous population. It is thus confusing that this factor was not apparently taken into consideration with the electrophysiology assessments.

This idea has been extensively addressed in our previous work (Nguyen et al., *Neuron* 2021), and indeed forms the basis for our interventions here. In Nguyen et al., we showed that inhibited neurons primarily belong to the VTA-AMg pathway (>85%) while activated neurons predominantly belong to the VTA-NAc pathway (>90%). In the current manuscript, dopaminergic neuron responses to nicotine were likewise separated according to their distribution (activated or inhibited) in order to account for DA neuron heterogeneity. Finally, this heterogeneity was not only taken into consideration but exploited in the design of our rescue experiments: because we saw changes in activation, but not in inhibition, of DA neurons by nicotine, our DREADD experiments focused only on the VTA-NAc pathway and were able to reverse the changes in anxiogenic response to nicotine seen after adolescent nicotine exposure.

We have added the following to the text to the manuscript to further clarify this important point.

Lines 196-198: Interestingly, mice exposed to nicotine in adolescence showed a stronger, adolescent-like, activation of DA neurons in response nicotine than their SAC treated counterparts, with no change in the magnitude of inhibition (Fig 3J). **This result suggests that there is greater activation in the VTA-NAc pathway in the NIC pre-treated mice.**

Lines 269-276: **With regards to nicotine response, we have shown that VTA-NAc projecting neurons are activated by nicotine, while VTA-Amg projecting neurons are inhibited by nicotine in adult mice, leading, respectively, to the expression of the reinforcing and anxiogenic properties of the drug²³.** How exactly adolescent experiences shape dopaminergic function within these distinct circuits is still an open question, with possibilities including to advance, slow, or misroute normal maturational processes. Here, our evidence suggests that **experience with nicotine in adolescence arrests the development of VTA-NAc and VTA-AMg DA circuits, keeping them in a prolonged adolescent state.**

REVIEWERS' COMMENTS

Reviewer #1 (Remarks to the Author):

I thank the authors for responding to my comments and concerns, and for making additional electrophysiology experimtns.

I am satisfied with their responses and how the authors improved the manuscript.

Reviewer #2 (Remarks to the Author):

The authors have been very responsive to the previous round of comments and have added new data and revisions that in my opinion, greatly strengthen the paper. I have no further critiques.

REVIEWERS' COMMENTS

Reviewer #1 (Remarks to the Author):

I thank the authors for responding to my comments and concerns, and for making additional electrophysiology experiments.

I am satisfied with their responses and how the authors improved the manuscript.

We thank the reviewer for their time and their remarks, which have improved the quality of the manuscript.

Reviewer #2 (Remarks to the Author):

The authors have been very responsive to the previous round of comments and have added new data and revisions that in my opinion, greatly strengthen the paper. I have no further critiques.

We thank the reviewer for their time and their remarks, which have improved the quality of the manuscript.